# Mapping the Timescale Organization of Neural Language Models

**Hsiang-Yun Sherry Chien, Jinhan Zhang & Christopher. J. Honey**
Department of Psychological and Brain Sciences
Johns Hopkins University
Baltimore, MD 21218, USA
`{sherry.chien,jzhan205,chris.honey}@jhu.edu`

## Abstract

In the human brain, sequences of language input are processed within a distributed and hierarchical architecture, in which higher stages of processing encode contextual information over longer timescales. In contrast, in recurrent neural networks which perform natural language processing, we know little about how the multiple timescales of contextual information are functionally organized. Therefore, we applied tools developed in neuroscience to map the "processing timescales" of individual units within a word-level LSTM language model. This timescale-mapping method assigned long timescales to units previously found to track long-range syntactic dependencies. Additionally, the mapping revealed a small subset of the network (less than 15% of units) with long timescales and whose function had not previously been explored. We next probed the functional organization of the network by examining the relationship between the processing timescale of units and their network connectivity. We identified two classes of long-timescale units: "controller" units composed a densely interconnected subnetwork and strongly projected to the rest of the network, while "integrator" units showed the longest timescales in the network, and expressed projection profiles closer to the mean projection profile. Ablating integrator and controller units affected model performance at different positions within a sentence, suggesting distinctive functions of these two sets of units. Finally, we tested the generalization of these results to a character-level LSTM model and models with different architectures. In summary, we demonstrated a model-free technique for mapping the timescale organization in recurrent neural networks, and we applied this method to reveal the timescale and functional organization of neural language models.[1]

## 1 Introduction

Language processing requires tracking information over multiple timescales. To be able to predict the final word "timescales" in the previous sentence, one must consider both the short-range context (e.g. the adjective "multiple") and the long-range context (e.g. the subject "language processing"). How do humans and neural language models encode such multi-scale context information? Neuroscientists have developed methods to study how the human brain encodes information over multiple timescales during sequence processing. By parametrically varying the timescale of intact context, and measuring the resultant changes in the neural response, a series of studies (Lerner et al., 2011; Xu et al., 2005; Honey et al., 2012) showed that higher-order regions are more sensitive to long-range context change than lower-order sensory regions. These studies indicate the existence of a "hierarchy of processing timescales" in the human brain. More recently, Chien & Honey (2020) used a time-resolved method to investigate how the brain builds a shared representation, when two groups of people processed the same narrative segment preceded by different contexts. By directly mapping the time required for individual brain regions to converge on a shared representation in response to shared input, we confirmed that higher-order regions take longer to build a shared representation. Altogether, these and other lines of investigation suggest that sequence processing in the

---

[1]The code and dataset to reproduce the experiment can be found at `https://github.com/sherrychien/LSTM_timescales`

brain is supported by a distributed and hierarchical structure: sensory regions have short processing timescales and are primarily influenced by the current input and its short-range context, while higher-order cortical regions have longer timescales and track longer-range dependencies (Hasson et al., 2015; Honey et al., 2012; Chien & Honey, 2020; Lerner et al., 2011; Baldassano et al., 2017; Runyan et al., 2017; Fuster, 1997).

How are processing timescales organized within recurrent neural networks (RNNs) trained to perform natural language processing? Long short-term memory networks (LSTMs) (Hochreiter & Schmidhuber, 1997) have been widely investigated in terms of their ability to successfully solve sequential prediction tasks. However, long-range dependencies have usually been studied with respect to a particular linguistic function (e.g. subject-verb number agreement, Linzen et al. 2016; Gulordava et al. 2018; Lakretz et al. 2019), and there has been less attention on the broader question of how sensitivity to prior context – broadly construed – is functionally organized within these RNNs. Therefore, drawing on prior work in the neuroscience literature, here we demonstrate a model-free approach to mapping processing timescale in RNNs. We focused on existing language models that were trained to predict upcoming tokens at the word level (Gulordava et al., 2018) and at the character level (Hahn & Baroni, 2019). The timescale organization of these two models both revealed that the higher layers of LSTM language models contained a small subset of units which exhibit long-range sequence dependencies; this subset includes previously reported units (e.g. a "syntax" unit, Lakretz et al., 2019) as well as previously unreported units.

After mapping the timescales of individual units, we asked: does the processing timescales of each unit in the network relate to its functional role, as measured by its connectivity? The question is motivated by neuroscience studies which have shown that in the human brain, higher-degree nodes tend to exhibit slower dynamics and longer context dependence than lower-degree nodes (Baria et al., 2013). More generally, the primate brain exhibits a core periphery structure in which a relatively small number of "higher order" and high-degree regions (in the prefrontal cortex, in default-mode regions and in so-called "limbic" zones) maintain a large number of connections with one another, and exert a powerful influence over large-scale cortical dynamics (Hagmann et al., 2008; Mesulam, 1998; Gu et al., 2015). Inspired by the relationships between timescales and network structure in the brain, we set out to test corresponding hypotheses in RNNs: (1) Do units with longer-timescales tend to have higher degree in neural language models? and (2) Do neural language models also exhibit a "core network" composed of functionally influential high-degree units? Using an exploratory network-theoretic approach, we found that units with longer timescales tend to have more projections to other units. Furthermore, we identified a set of medium-to-long timescale "controller" units which exhibit distinct and strong projections to control the state of other units, and a set of long-timescale "integrator units" which showed influence on predicting words where the long context is relevant. In summary, these findings advance our understanding of the timescale distribution and functional organization of LSTM language models, and provide a method for identifying important units representing long-range contextual information in RNNs.

## 2 RELATED WORK

**Linguistic Context in LSTMs**. How do LSTMs encode linguistic context at multiple timescales? Prior work suggested that the units sensitive to information that requires long-range dependencies are sparse. By ablating one unit at a time, Lakretz et al. (2019) found two units that encode information required for processing long-range subject-verb number agreement (one for singular and one for plural information encoding). They further identified several long-range "syntax units" whose activation was associated with syntactic tree-depth. Overall, Lakretz et al. (2019) suggests that a sparse subset of units tracks long-range dependencies related to subject-verb agreement and syntax. If this pattern is general – i.e. if there are very few nodes tracking long-range dependencies in general – this may limit the capacity of the models to process long sentences with high complexity, for reasons similar to those that may limit human sentence processing (Lakretz et al., 2020). To test whether long-range nodes are sparse in general, we require a model-free approach for mapping the context dependencies of every unit in the language network.

**Whole-network context dependence**. Previous work by Khandelwal et al. (2018) investigated the duration of prior context that LSTM language models use to support word prediction. Context-dependence was measured by permuting the order of words preceding the preserved context, and

observing the increase in model perplexity when the preserved context gets shorter. Khandelwal et al. (2018) found that up to 200 word-tokens of prior context were relevant to the model perplexity, but that the precise ordering of words only mattered within the most recent 50 tokens. The token-based context-permutation method employed in this study was analogous to the approach used to measure context-dependence in human brain responses to movies (Hasson et al., 2008) and to auditory narratives (Lerner et al., 2011).

Inspired by the findings of Khandelwal et al. (2018) and Lakretz et al. (2019), in the present study we set out to map the context-dependence across all of the individual units in the LSTM model. This enabled us to relate the timescales to the effects of node-specific ablation and the network architecture itself. In addition, our context manipulations included both context-swapping (substituting alternative meaningful contexts) and context-shuffling (permuting the words in the prior context to disrupt inter-word structure), which allowed us to better understand how individual words and syntactically structured word-sequences contribute to the the context representation of individual hidden units.

## 3 METHODS

### 3.1 LANGUAGE MODELS AND CORPUS

We evaluated the internal representations generated by a pre-trained word-level LSTM language model (WLSTM, Gulordava et al., 2018) as well as a pre-trained character-level LSTM model (CLSTM, Hahn & Baroni, 2019) as they processed sentences sampled from the 427804-word (1965719-character) novel corpus: *Anna Karenina* by Leo Tolstoy (Tolstoy, 2016), translated from Russian to English by Constance Garnett.

For the WLSTM, we used the model made available by Gulordava et al. (2018). The WLSTM has a 650-dimensional embedding layer, two 650-dimensional hidden layers and an output layer with vocabulary size 50,000. The model was trained and tested on Wikipedia sentences and was not fine-tuned to the novel corpus. Therefore, we only used sentences with low perplexity from the novel in our main timescale analysis. We performed the same analysis using the Wikipedia test set from Gulordava et al. (2018) and obtained similar results (See Section 5.3, Figure A.4A, Appendix A.2.1). For the CLSTM, we used the model made available by Hahn & Baroni (2019). The CLSTM has a 200-dimensional embedding layer, three 1024-dimensional hidden layers and an output layer with vocabulary size 63. The model was trained on Wikipedia data with all characters lower-cased and whitespace removed. We tested the model with sentences sampled from *Anna Karenina* as the WLSTM model, and we obtained bits-per-character (BPC) similar to what Hahn & Baroni (2019) reported in their original work.

### 3.2 TEMPORAL CONTEXT CONSTRUCTION PARADIGM

In order to determine the processing timescales of cell state vectors and individual units, we modified the "temporal context construction" method developed by Chien & Honey (2020). Thus, the internal representations of the model were compared across two conditions: (1) the Intact Context condition and (2) the Random Context condition. In both conditions, the model was processing the same shared sequence of words (for example, segment B), but the preceding sentence differed across the two conditions. In the Intact Context condition, the model processed segment B (the shared segment) preceded by segment A, which was the actual preceding context from the original text. In the current study, for example, segment A and B are connected by ", and" within long sentences from the novel corpus (Figure 1A), to ensure the temporal dependencies between A and B. In the Random Context condition, however, the model processed the same shared input (segment B), but the context was replaced by segment X, which was a randomly sampled context segment from the rest of the corpus. Segment X was therefore not usually coherently related to segment B. For the WLSTM timescale analysis, we chose long sentences in the Intact Context condition that satisfied the following constraints: (1) mean perplexity across all words in the sentence $< 200$, (2) the shared segment was longer than 25 words, and (3) the context segment was longer than 10 words. 77 sentences are included as trials in our analyses. In the Random Context condition, we preserved the same shared segments and randomly sampled 30 context segments (each longer than 10 words) from other parts of the novel. For the CLSTM timescale analysis, we used the same 77 long sentences in the Intact

Context condition, and randomly sampled 25 context segments (with length > 33 characters) for the Random Context condition.

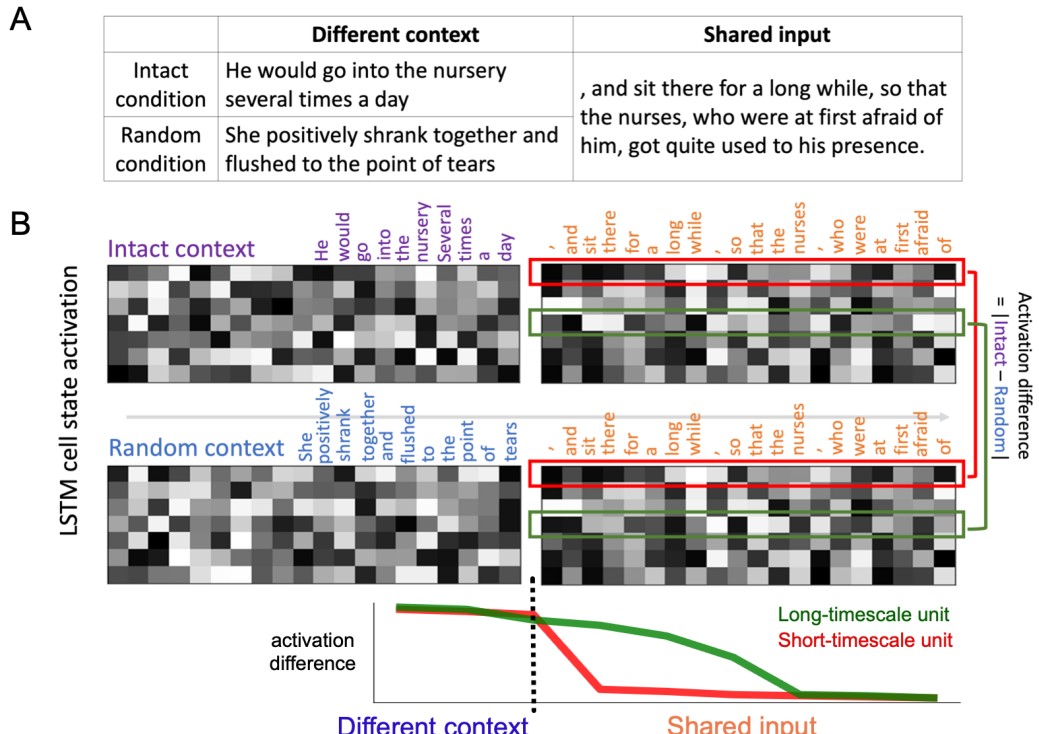

Figure 1: Method for mapping processing timescales of individual units. **A.** Example sentences for the model to process in the Intact Context and Random Context condition. In the Intact Context condition, the shared segment is preceded by an intact context from the corpus; while in the Random Context condition, this preceding context segment is replaced by randomly sampled context segments. **B.** Schematic hidden state activation of the neural network. When the model starts to process the shared segment preceded by different context between the two context conditions, the hidden unit activation difference (i.e. the mean absolute difference of unit activation between the two conditions) decreases over time with different rates. The expected decreasing pattern of activation difference of a long-timescale unit and a short-timescale unit are shown schematically in the green and red curves, respectively.

In brief, the model is processing the same input (the shared segment) with different preceding context (the intact vs. random context). We can now measure the context dependence of individual units by examining how the cell state activations differ between the two conditions, while the network is processing the shared segments with identical input. Any difference in internal representations must arise from the context manipulation, since the current input is the same. A decrease in activation difference over time implies that the units exposed in the Intact context and Random context start to build a similar representation as they process the shared input. For a long-timescale unit, whose current state is dependent on information in the far-preceding context, we will see that the activation difference is preserved across contexts (Figure 1B, green curve), even while the unit is processing the shared input. On the other hand, for a short-timescale unit whose activation is driven largely by the current input, we will see that the activation difference drops quickly (Figure 1B, red curve) as the unit processes the shared input.

## 4 HIERARCHICAL ORGANIZATION OF TIMESCALES ACROSS LAYERS

Do higher levels of the LSTM model exhibit greater context-dependence? Lakretz et al. (2019) observed that long-range functional units were more common in higher layers, and in general, higher-levels of hierarchical language model exhibit longer range context-dependence (Jain et al., 2019; Jain & Huth, 2018). Therefore, to validate our stimuli and the sensitivity of our methods, we first compared the processing timescales of different hidden layers in both of the LSTMs, by correlating the cell state vectors, column by column, between the Intact condition and Random condition.

We found that both layers showed near-zero correlation when processing the different context, and the correlation increased as they began to process the shared input. In the WLSTM, the correlation increased more slowly for second-level cell state vectors than for first-level cell state vectors. Thus, the representation of second-level cell state is more sensitive to the different context than the first level. Similarly, for the CLSTM model, the third-level cell state exhibited longer-lasting context sensitivity than lower levels (Figure 2). This observation of longer context-dependence in higher stages of processing is consistent with prior machine learning analyses (Lakretz et al., 2019; Jain & Huth, 2018) and is also analogous to what is seen in the human brain (Hasson et al., 2015; Chien & Honey, 2020; Lerner et al., 2011; Jain et al., 2019). Based on the finding of longer context dependence in higher layers, we examined single units in the highest level hidden units, i.e. the second level of WLSTM (n=650) and the third level of CLSTM (n=1024).

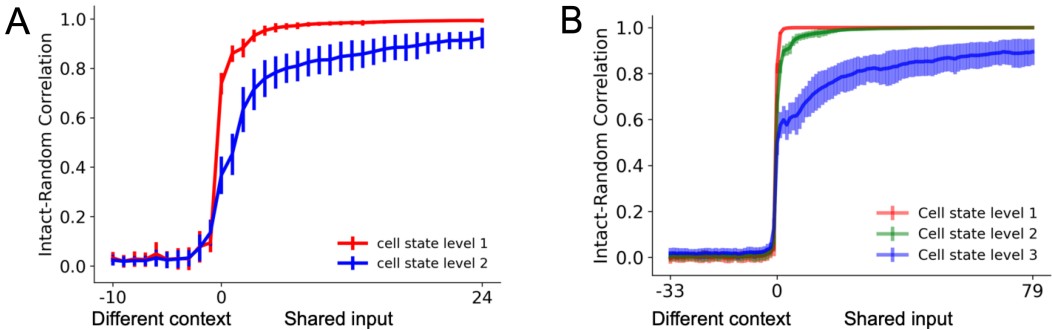

Figure 2: Context effect measured by cell-state vector correlation at different layers in word-level LSTM (WLSTM) and character-level LSTM (CLSTM). **A.** Correlation curves of the WLSTM cell-state vectors across the Intact Context condition and Random Context condition as a function of input token. In both models, the correlation increased as the models began to process the shared segment. Higher-level cell states exhibited a slower increase in correlation, compared to lower-level cell states, indicating that the higher-levels retain more of the prior context information for longer. **B.** As for A, but applied to the three levels of CLSTM. Similar to the WLSTM, higher-level cell state of the CLSTM showed more context sensitivity than the lower-level cell state.

## 5 PROCESSING TIMESCALES OF INDIVIDUAL UNITS WITHIN LSTM LAYERS

### 5.1 QUANTIFYING SINGLE UNIT TIMESCALES

We examined the absolute single unit activation difference when processing the shared segments preceded by different context. As expected, most of the hidden units showed different activation when the input tokens were different (i.e. while processing the non-shared context in the Intact Context and Random Context conditions). However, once the shared input tokens begin (at $t = 0$) the Intact-Random activation differences drop (Figure A.1A, A.1B).

We used the rate at which the curves drop to quantify the processing timescale, as this is a measure of how quickly the responses align across different context conditions. To quantify the timescale of individual units, we fit the activation difference curves with a logistic function:

$$Y(x) = \frac{L}{1 + e^{-k(x-x_0)}} + d \tag{1}$$

As shown in Figure A.1A and Figure A.1B, the logistic function fit the raw activation difference curves. We then computed the "timescale" of each unit as the time-to-half-maximum of the logistic decay. In particular, for the WLSTM we used the activation difference $Y(0)$ at the beginning of the shared segment, and at the end of the shared segment $Y(24)$ ($Y(79)$ for the CSLTM) to calculate the time-to-half-maximum of unit $i$ as:

$$\text{timescale}_i = \lceil Y^{-1}(\frac{Y_i(0) - Y_i(24)}{2}) \rceil \tag{2}$$

where the inverse function $Y^{-1}(y)$ identifies the largest integer $t$, for which $Y(t) < y$. We included 635 units in WLSTM and 1012 units in CLSTM for further analysis after excluding the units which could not be accurately fit by a logistic function (See Appendix A.1).

## 5.2 Distribution of Unit Timescales in LSTM Language Models

The results showed that of the 635 WLSTM units whose processing timescale we mapped, approximately 70% of the units were insensitive to long-range context (processing timescale $< 3$ words): their activation difference dropped immediately at onset of the shared segment. In contrast, only approximately 13% of the units had a timescales $> 7$ words (Figure A.2A). Figure 3A shows the absolute activation difference of all units in WLSTM sorted by timescale (long to short). Some of the longer-timescale units continued to exhibit a large activation difference even when processing the shared segments for more than 20 tokens.

As we were testing the same word-level LSTM previously studied by Lakretz et al. (2019), we began by examining the timescales of hidden-state units that were already known to be involved in processing context-dependence language information: a "singular number unit" 988, a "plural number unit" 776, and a "syntax unit" 1150. We found that, compared to other units, both "number" units had medium timescales ($\sim 3$ words, ranked 129 of 635 units), while the "syntax" unit had a long timescale ($\sim 7$ words, ranked 64 of 635 units) (Figure A.1).

We repeated the timescale mapping in the CLSTM model, and again identified a small subset of long-timescale units (Figure 3B, Figure A.2B). Although there were overall more units in CLSTM, over 63% of the units were insensitive to the context (timescale $< 3$ characters). Fewer than 15% of the units exhibited timescale $> 10$ characters, and the unit with the longest timescale only dropped to its half-maximum activation-difference after 50 characters of shared input.

## 5.3 Timescale Variation across Datasets and Context Conditions

To ensure that the timescales we measured were robust across datasets, we conducted the same analysis on WLSTM using the Wikipedia testing dataset used in Gulordava et al. (2018). The mapped timescales were highly correlated (r=0.82, p<0.001) across the *Anna Karenina* dataset and the Wikipedia dataset (Appendix A.2.1, Figure A.4A).

Similarly, to confirm that the timescales measured were not specific to our testing using the ", and" conjunction point, we also measured timescales at an alternative segmentation point, and found that the timescales were largely preserved (r=0.83, p<0.001), notwithstanding there were a small set of notable exceptions (Appendix A.2.2, Figure A.4B).

Although we measured the timescales of context dependence using "token distance", these measures are not invariant to changes in the the "syntactic distance". For example, if one were to replace a comma with a "full stop", then the token distance would be unaltered but the syntactic distance could be greatly altered. Indeed, we found that most units showed little context dependence when the preceding context segment ended with a "full stop", which served as a clear signal for the end of a sentence (Appendix A.2.3, Figure A.4C).

Finally, we examined whether the contextual information retained by the language models (and the associated timescales measurement) was sensitive to linguistic structure in the context, or whether it was primarily driven simply by the presence or absence of individual words. To this end, we generated text for the Random Context condition by shuffling the order of words from the Intact segment. We found that while the presence of individual words did play an important role in determining the context representations (and thus the timescales), several units showed a longer timescale when the prior context was composed of coherently structured language (Appendix A.2.4, Figure A.4D).

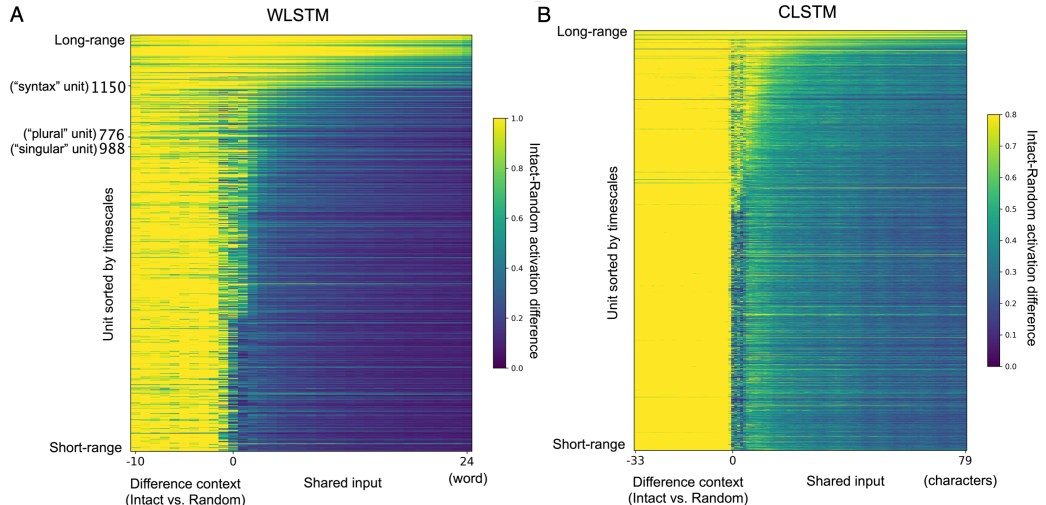

Figure 3: Timescale organization in word-level LSTM (WLSTM) and character-level LSTM (CLSTM) language model. **A.** Absolute activation difference for each WLSTM hidden unit over time, with units (rows) sorted by timescales. A small set of long-timescale units (top) sustain an activation difference during shared segment processing, but most (bottom) are context-insensitive short-timescale units. **B.** Absolute activation difference for each CLSTM unit over time, with units sorted by timescales. Similar to the WLSTM, a small set of long-timescale CLSTM hidden units maintain long-range contextual information.

## 6 CONNECTIVITY OF MEDIUM- TO LONG-TIMESCALES UNITS IN LSTMS

Having mapped the timescales of each processing unit, we next asked: how does the processing timescale of a unit relate to its functional role within the network? More specifically, are units with longer timescales also units with high degree in the connectivity network? To answer these questions, we analyzed (1) the projection strength of each unit and (2) the similarity of the overall projection pattern (hidden-to-gates) across different units. The projection patterns were defined using the direct weight projections from one hidden unit at time $t$ to the input and forget gate of other hidden units at time $t + 1$.

In LSTMs, the amount of contextual ($c_{t-1}$) and input ($\tilde{c}_t$) information stored in the cell state ($c_t$) is determined by the forget gate ($f_t$) and input gate ($i_t$) activation (Eq. 3); and the activation of the gates $i_t$ and $f_t$ are determined by the current input at time $t$ and the hidden units at time $t-1$ through weight matrices $U$ and $W$ (Eq. 4, 5).

$$c_t = f_t \odot c_{t-1} + i_t \odot \tilde{c}_t \tag{3}$$

$$i_t = \sigma(U_i x_t + W_i h_{t-1} + b_i) \tag{4}$$

$$f_t = \sigma(U_f x_t + W_f h_{t-1} + b_f) \tag{5}$$

Here, we were interested in understanding how the contextual information over different timescales is projected from the hidden units to the input and forget gates of other units, and further influence the update of cell states. Thus, we analyzed the network connectivity focusing on the weight matrices $W_i$ and $W_f$ within the highest layer of the WLSTM or CLSTM.

### 6.1 STRONG PROJECTIONS FROM LONG-TIMESCALE HIDDEN UNITS TO GATE UNITS

Units with longer processing timescales made a larger number of strong projections ($|\text{z-score}| >$ 5, Appendix A.3) to the input and forget gates of other units in both WLSTM (r=0.31, p<0.001, Figure 4A) and CLSTM models (r=0.24, p<0.001, Figure A.5A). Furthermore, we found that the "syntax" unit (Unit 1150) reported by Lakretz et al. (2019) in the WLSTM model possessed the largest number of strong projections to the input and forget gates of all other units, and the major recipients from Unit 1150 were units with medium- to long-timescale units (Figure 4B).

## 6.2 IDENTIFY CONTROLLER UNITS IN LSTM LANGUAGE MODELS

The presence of strong projections from the "syntax" unit to other long-timescale units motivated us to further explore whether high-degree, long-timescale units in the LSTM also densely interconnect to form a "core network", perhaps analogous to what is seen in the brain (Hagmann et al., 2008; Mesulam, 1998; Baria et al., 2013). If so, this set of units may have an especially important role in controlling how prior context is updated and how it is used to gate current processing, analogous to the controller system in the brain (Gu et al., 2015). To identify these putative "controller units", we binarized the network by identifying the top 258 projection weights from the weight matrices (see Appendix A.3), which provided the edges for a network analysis. We then used k-core analysis (Batagelj & Zaversnik, 2003) to identify the "main network core" (the core with the largest degree) of the network (Figure A.3). At the maximal $k = 5$, the k-core analysis yielded a set of densely interconnected nodes, composed of many long-timescale and medium-timescale units (Figure A.3), also labeled in red in Figure 4A. We (tentatively) refer to this set as the "controller" set of the network. Performing the same k-core analyses on the CLSTM model, we observed that the main core network was again composed of disproportionately many medium and long-timescale "controller" units (Figure A.5A).

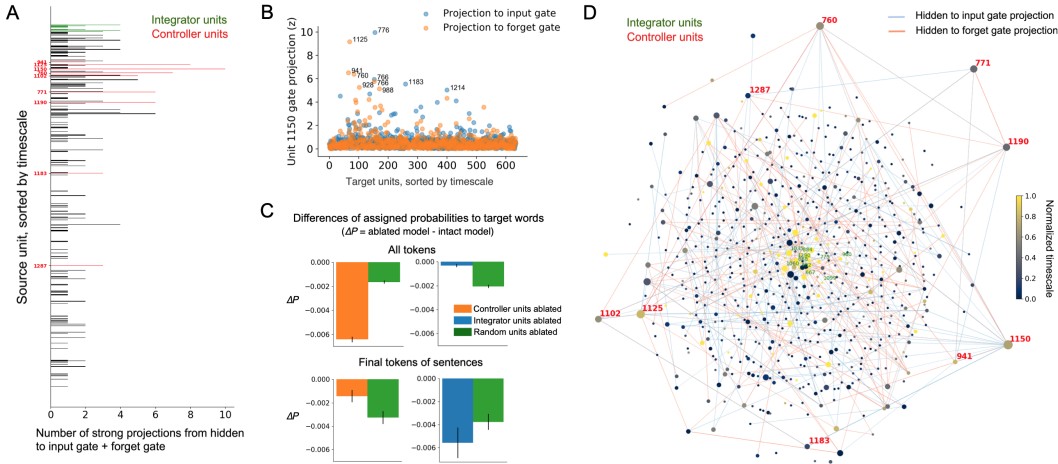

Figure 4: Timescale and connectivity organization in a word-level LSTM. **A.** Long-timescale units exhibited stronger projections from the hidden state at time $t$ to the forget gate and input gate at time $t + 1$. **B.** Strength of hidden-forget gate and hidden-input gate projections from a high-degree "syntax" unit to all other units. The units receiving strong projections (|z-score|$> 5$) are labeled. **C.** Ablating the two sets of long-timescale units results in different impact to the LSTM performance. Specifically, ablating "controller" units impaired overall word prediction (upper panel), while ablating "integrator" units impaired prediction of words in the later part of the sentences (bottom panel). **D.** Multi-dimensional scaling representation of network connectivity. The distance between two nodes indicates the similarity of their hidden-to-gate connection patterns. The size of each node indicates its degree (the number of strong projections from that node to the gate units). An edge between nodes indicates a significant hidden-to-gate projection between them.

## 6.3 DISTINCTIVE ROLES OF LONG-TIMESCALE CONTROLLER AND INTEGRATOR UNITS

We used multi-dimensional scaling (MDS) to visualize the similarity of projection patterns across LSTM units. We recovered a 2-dimensional MDS embedding, in which the inter-unit distances was defined based on the similarity of their hidden-to-gate projection patterns (i.e., similarity of values in the unthresholded LSTM weight matrices $W_i$ and $W_f$). We visualized the MDS solution as a graph structure, in which each node is a unit, and the edges reflect connection properties of that unit. Figure 4D shows the resulting 2-D space, with units color-coded by their timescale.

"Controller units" (labeled on Figure 4D) were positioned around the periphery of the MDS space, suggesting that these units expressed projection patterns that were distinct from other "controller" units and also from the rest of the network. In contrast, we observed several long-timescale units

positioned in the center of the MDS space, suggesting that the projection patterns of these units were similar to the mean projection pattern. We refer to this more MDS-central set as the "integrator units" (labeled in green in Figure 4A). Similar to the WLSTM, the projection patterns of the "controller units" in the CLSTM were distinct from other units in the network, according to the MDS results (Figure A.5C). However, we did not observe "integrator units" positioned in the center of the MDS space of the CLSTM.

Are the "controller" and "integrator" units particularly important for the model's ability to predict the next token? To test the functional importance of these subsets of units, we conducted group ablation analyses (See Appendix A.4). Ablating controller units reduced the accuracy of token prediction overall, while ablating integrator units only reduced prediction accuracy for the last words of the sentences (Figure 4C). The results confirm that the putative controller and integrator nodes are functionally significant, with distinctive roles in the WLSTM language model.

Finally, to test the generalization of the timescale and connectivity analyses to a different model architecture, we conducted preliminary analyses on a Gated Recurrent Unit (GRU) language model (Cho et al., 2014) and another word-level LSTM model with a smaller hidden size (100 units) per layer. The models were trained using similar parameter settings as in Gulordava et al. (2018) until they converged without any model-specific optimization. We found similar sparsity of long-timescale units in both models, but did not observe the same relationship between timescales and connectivity (Appendix A.5; A.6; Figure A.7; A.8; A.9; A.10).

## 7 DISCUSSION

We demonstrated a new method for mapping the timescale organization in recurrent neural language models. Using this method, we mapped the timescale distributions of units within word-level and character-level LSTM language models, and identified a small set of units with long timescales. We then used network analyses to understand the relationship between the timescale of a unit and its connectivity profile, and we distinguished two subsets of long-timescale units with seemingly distinctive functions. Altogether, we proposed methods combining timescale and connectivity analyses for discovering timescale and functional organization in language models.

The units with longer processing timescales included some units whose role in long-range language dependencies had already been established (Lakretz et al., 2019), but almost all of the long timescale units are of unknown function. The timescale mapping procedure described here provides a model-free method for identifying nodes necessary for long-range linguistic and discursive processes (e.g. tracking whether a series of words constitutes an assertion or a question). Future studies of these neural language models could focus on the specific linguistic information tracked by the long-timescale units, especially the "controller" units which control the information flow of other units in the network.

The current study measured unit timescales using a simple token distance, and so the method may be applied to understanding recurrent neural nets beyond language models. It will be insightful for future studies to investigate whether the processing timescales characterized via token distance are comparable to those measured using functional measures, such as syntactic distance. Relatedly, while we explored the timescale variance under several context conditions, a more thorough investigation will be needed to examine how the timescales of individual units may vary at different positions within a sentence, both in terms of token location and syntactic location.

Processing timescales may exhibit an analogous hierarchical organization in LSTMs and in the human cerebral cortex: in both cases, a subset of nodes with high degree and high inter-connectivity express unusually long timescales. More detailed testing of this apparent correspondence is required, however, because units within an LSTM layer are not spatially embedded and constrained as in biological brains, and thus the LSTM units do not express a spatially graded timescale topography.

ACKNOWLEDGMENTS

C.J.H and H-Y.S.C gratefully acknowledge the support of the National Institutes of Mental Health (grant R01MH119099)

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

# A  APPENDIX

## A.1  UNITS EXCLUDED FROM TIMESCALE ANALYSIS

We excluded 1 unit in the WLSTM model and 5 units in CLSTM model which were not properly fit using the logistic function; we further excluded 14 units in the WLSTM model and 7 units in the CLSTM model which either did not show a non-zero activation difference before the shared segment started, or whose activation differences increased when started to process the shared segment. After these exclusions, 635 units remained in the WLSTM and 1012 units remained in the CLSTM for further analysis.

## A.2  TIMESCALE ANALYSES ACROSS DIFFERENT DATASETS AND CONTEXT CONDITIONS

### A.2.1  WIKIPEDIA TEST DATASET

The *Anna Karenina* corpus used in the current study has a different linguistic structure from the Wikipedia corpus on which the WLSTM and CLSTM models were trained. Although we analyzed only the *Anna Karenina* sentences with low perplexity, it was important to test the robustness of our results across datasets. Thus, we mapped the timescale of each unit using the Wikipedia test set, as used by Gulordava et al. (2018). Specifically, we sampled 500 long sentences containing ", and" for the Intact Context condition. As before, we generated sentences by preceding the "shared input" segment (after the conjunction) with either the original prior context segment, or a randomly chosen prior context segment. Same as the original analysis, we then replaced the context segment with 30 context segments randomly sampled from other parts of the test set for generating the Random Context condition. The mapped timescales using the Wikipedia test set were highly correlated with the novel corpus, suggesting the robustness of unit timescales (Figure A.4A).

### A.2.2  TIMESCALES MEASURED IN THE MIDDLE OF A SENTENCE

To examine how the timescales of individual units may vary across different positions in a sentence, we varied the location of the segmentation point. Instead of using the conjunction (", and") as the segmentation point, we chose an arbitrary segmentation point: the 15th token of a long sentence, to separate context segment and shared input segment. In the Random Context condition, we replaced the context segment with the first 15 tokens from other sentences of the corpus. We found that the unit timescales were highly correlated with the condition where we used the conjunction as the segmentation point with several units shift their timescales to either directions (Figure A.4B). This analysis was conducted using Wikipedia test set.

### A.2.3  TIMESCALE RESET AT THE BEGINNING OF A SENTENCE

To examine if the timescales of individual units can flexibly reset at the beginning of a sentence, we conducted the same timescale analysis but using a "full stop" as the segmentation point instead of the conjunction ", and". Thus, if the original test string was "The girl kicked the call, and the boy caught it", then the full-stop version of the test string would be "The girl kicked the ball. The boy caught it." In this setting, the context segment and shared input segment in the Intact Context condition are two consecutive sentences. To ensure the temporal dependence between the context segment and shared input segment, we sampled 100 consecutive sentence pairs from the *Anna Karenina* corpus. Note that this is not possible using the Wikipedia test set from Gulordava et al. (2018), because that set is composed of unrelated sentences. The Random Context condition was generated by replacing the first sentence with randomly sampled sentences from other parts of the novel. We found that when using "full stop" to segment context and shared input, most units in the network showed timescale near 0, indicating near-zero dependence on the linguistic context from the text preceding the full stop (Figure A.4C). This suggests that the units in LSTM tend to "reset" their context representation at the beginning of a sentence.

### A.2.4  CONTEXT REPRESENTATION SHAPED BY INDIVIDUAL WORDS

Inspired by the token-shuffling procedure of Khandelwal et al. (2018), we explored whether the context representations of individual units in the LSTM were shaped by individual words, rather

than coherent sequences of words. For this analysis, instead of replacing the context with syntactically structured segments from other part of the corpus, we generated the "random context" by shuffling the order of words within the context segment. We then mapped the unit timescales as before, by examining the unit activation difference as a function of the distance from the onset of shared input. Intriguingly, we found that most of the units showed similar timescales across the context-replacement and context-shuffling procedures (Figure A.4D). This suggests that the context representations in LSTMs largely depend on the *presence* of individual words in the context, rather than their appearance within coherent linguistic sequences. However, we did observe a subset of units (labeled in the Figure, and almost all long-timescale units) whose timescales were longer when context was replaced rather than shuffled. For this subset of units, the ability to maintain a representation of prior context over many tokens depends on that prior context being a coherent linguistic sequence. This subset of units are a promising target for future studies of syntactic representations in LSTMs.

### A.3 IDENTIFYING STRONG HIDDEN-TO-GATE PROJECTIONS

First, for each hidden unit, we concatenated the corresponding rows in the $W_{hi}$ and $W_{hf}$ matrices, to generate a single "hidden-to-gate" projection vector for that hidden unit. Next we we z-scored the vector to get standardized projection values from that unit to all other units in the network. Using |z-score|$> 5$ as criterion, we identified a total of 258 "strong projections" from all hidden units to the input gate and forget gate in the WLSTM. The projection strength of each unit was then calculated based on its number of "strong projections" (Figure 4A). Although the criterion |z-score|$>$ was selected to better visualize the results in Figure 4, different criteria did not change the results that units with longer timescales have more strong projections. For example, using |z-score|$> 3$ as threshold we obtained corr(timescale, projections) = 0.30, p<0.001; |z-score|$> 4$ we obtained corr(timescale, projections) = 0.35, p<0.001.

Next, we identified the edges corresponding to the top 258 magnitude weight-values within the combined $W_{hi}$ and $W_{hf}$ matrices. Together, these edges formed a "strong-projection network". Finally, we used k-core analysis to identify the main core of the strong-projection network. This main core composed our "controller units" (Figure A.3).

Using the same criteria and method, we identified a total of 390 "strong projections" from all hidden units to the input gate and forget gate in the CLSTM. We then extracted the top 390 weight values from the weight matrices to construct a "strong-projection network" and again identified the main core network, composed the "controller units" for the CLSTM model (Figure A.5A, A.5B)

### A.4 ABLATION ANALYSES ON PUTATIVE CONTROLLER AND INTEGRATOR UNITS

To examine the non-trivial roles of the controller and integrator units identified in the word-level LSTM model, we performed a preliminary group ablation analysis to look at how ablating the controller units influences model performance on predicting the next token, relative to the ablation of a random set of units. Specifically, since long-timescale integrator units should have most effect predicting tokens at the later part of the sentences (i.e., when more context is integrated), we examined the model performance on predicting tokens at two different positions: (1) all the tokens regardless of their positions in the sentences ("All tokens" condition), and (2) the last tokens of sentences ("Final tokens" condition).

We evaluated the effects of ablation on model performance by measuring the differences of probabilities ($\Delta P$) assigned to the target words ($\Delta P$ = probability of target word in ablated model minus probability of target word in original model). Ablation effects for controller units ($N$=9) and integrator units ($N$=10) were compared against a baseline of ablating the same number of randomly-selected units from layer 2 of the LSTM (Figure 4C). We used the test corpus used by Gulordava et al. (2018) and measured the average performance of each model across 100 text-batches, randomly sampled from the Wikipedia test dataset. Each text-batch was composed of 1000 tokens that start at the beginning of a sentence.

In the "All tokens" condition, we calculated the $\Delta P$ for every token in the tested text, while in the "Final tokens" condition, we calculated $\Delta P$ only at the last token of every sentence (i.e. the token

right before the full stop"." of each sentence). We then average the $\Delta$P in both conditions across text-batches to get a mean performance difference between the ablated model and the intact model.

Ablating controller units reduced the probabilities assigned to the target words, more so than ablating random units (Figure 4C, controller vs. random across 100 text batches: Cohen's $d$ = -4.85, $t$ = -34.28, p<0.001). In contrast, ablating integrator units reduced the probabilities less than ablating random units (integrator vs. random: Cohen's $d$ = 2.50, $t$ = 17.67, p<0.001). We hypothesized that that the integrator units mostly influence the model's prediction performance for tokens where long-range information is especially relevant, such as in the later portions of clauses and sentences. Consistent with this, we found that, when we examined the ablation effects only for tokens in the final position of a sentence, ablating integrator units reduced the probabilities more than ablating random units (Cohen's $d$ = -0.34, $t$ = -2.41, p = 0.017). Interestingly, ablating controller units reduced the probability of sentence-final targets less than random units (Cohen's $d$ = 0.67, $t$ = 4.74, p<0.001).

In summary, these ablation results indicate a non-trivial functional role for the controller and integrator units, despite the fact that each subset of units is composed of only 10 amongst 650 total hidden units. Also, the putative controller and integrator sets appear to have distinctive roles within the WLSTM, with the controllers supporting accurate predictions overall, while the integrator units appear to boost accurate predictions at the end of sentences.

### A.5    MAPPING THE TIMESCALE ORGANIZATION IN A GRU LANGUAGE MODEL

#### A.5.1    TRAINING

To explore whether the timescale mapping methods, and our findings, may generalize to other model architectures, we trained and studied a word-level GRU language model (Cho et al., 2014). As far as possible, we applied similar parameters in the GRU as were used for the LSTM by Gulordava et al. (2018): the same Wikipedia training corpus, the same loss function (i.e. cross-entropy loss), and the same hyperparameters except for a learning rate initialized to 0.1, which we found more optimal to train the GRU. The GRU model also had two layers, with 650 hidden units in each layer.

We trained the GRU model for 30 epochs, at which point the GRU converged to a validation perplexity of 118.36. Note that since we adapted similar training settings as were used for training the LSTM model by Gulordava et al. without model-specific optimization, the perplexity is higher than that of the LSTM model reported in Gulordava et al. (2018) (perplexity = 52.1 in the English corpora, after training for 40 epochs and selecting the model with the lowest perplexity out of 68 combinations of different hyperparamters). We then analyzed the timescale of its hidden units using the same method as was used for analyzing the LSTMs, and using the test data derived from the training Wikipedia corpus.

#### A.5.2    TIMESCALE ORGANIZATION OF A GRU MODEL

Similar to the LSTM model of Gulordova et al, the majority of the units in the GRU also showed shorter timescales. More specifically, we found: (1) the second layer of the GRU model was more sensitive to prior context than the first layer, as in the LSTM (A.7A); (2) the distribution of timescales across units was similar in the GRU and LSTM, although the GRU showed a more right-skewed distribution with a larger proportion of short-timescale units (A.7B, C).

#### A.5.3    TIMESCALE VERSUS NETWORK CONNECTIVITY IN A GRU MODEL

We also performed the timescale vs. network connectivity analyses on the GRU model. Because the update of hidden states in GRU are controlled by the reset and update gate, we measured the projection patterns of hidden units by analyzing the matrix of combined hidden-to-update-gate and hidden-to-reset-gate weights. In contrast to the LSTM models, hidden units in the GRU that we trained did not show a relationship between longer timescales and stronger hidden-to-gate projections (A.8A). Moreover, when using k-core analysis to identify subunits of interconnected high-degree units, the core network in the GRU contained many units with long to short timescales. Interestingly, when we visualized the position of the k-core units in the MDS space, they tended to locate at the edge of the space, similar to what we found in LSTM. This indicates that, as in the LSTM, the core units

in the GRU have distinctive profiles, distant from one another and from other units in the network (A.8B). However, we did not observe the pattern of "integrator units" in the GRU as in the LSTM.

These apparent similarities and differences between LSTM and GRU are intriguing, but we emphasize that (1) the perplexity of this GRU model is much higher than the LSTM, due to the sub-optimal parameter settings, and that (2) comparing the LSTM and GRU connection patterns is not straightforward, as the overall distribution of weights is different. Further work will be required to determine comparable thresholds for "strong" projections and "high-degree units" in each case. As we noted in the manuscript and above, the connectivity results are exploratory; however, we believe that the GRU analysis demonstrates how these methods can be extended to map and compare the functional organization of language models of different architecture.

Finally, we note that when conducting the timescale analysis on an incompletely trained GRU model (trained $\sim$10 epochs, validation perplexity $\approx$ 350), the timescale distribution was more right-skewed (Figure A.6B) than the better-trained GRU (Figure A.7B). Altogether, these results suggest that the long-timescale units in GRU were gradually formed during the training process.

## A.6 MAPPING THE TIMESCALE ORGANIZATION IN A WORD-LEVEL LSTM WITH DIFFERENT HIDDEN SIZE

To examine whether the number of hidden units in the model would affect the timescale organization in an LSTM, we trained another 2-layer word-level LSTM model with the same Wikipedia corpus and similar parameter settings as in Gulordava et al. (2018), but with only 100 hidden units in each layer. We called this model LSTM-100. We trained the model for 56 epochs until the model converged to a validation perplexity 98.75, and conducted the same analysis as described in the main text to map the timescales of LSTM-100. Because LSTM-100 have overall less weight connections, we use |z-score|> 3 as criteria to determine the "strong" hidden-to-gate projections for connectivity analyses.

Regarding the timescale distribution in LSTM-100, we found that the results were similar to the 650-unit word-level LSTM model, in that: (1) the second layer of LSTM-100 showed more context sensitivity than the first layer, and (2) although it was difficult to quantitatively compare the unit-level timescale distribution between the LSTM-100 model and the LSTM with 650 units, they both contain a similarly small subset of long-timescale units. (A.9).

We did not observe a significant correlation between the unit timescale and number of strong projections generated by each unit in the LSTM-100 model: the long-timescale units in the LSTM-100 did not have more connections than short-timescale units. When visualizing the MDS space of connectivity similarity of LSTM-100, the "controller units" identified using the k-core analysis were located in the edge of the space, similar to the 650-unit LSTM model. Interestingly, we observed a subset of long-timescale units in the center of the MDS space, analogous to the "integrator units" found in the 650-unit LSTM model. Altogether, the pattern of "integrator units" might be a commonly evolved feature that is shared between LSTM model architectures, but not with GRU architectures.

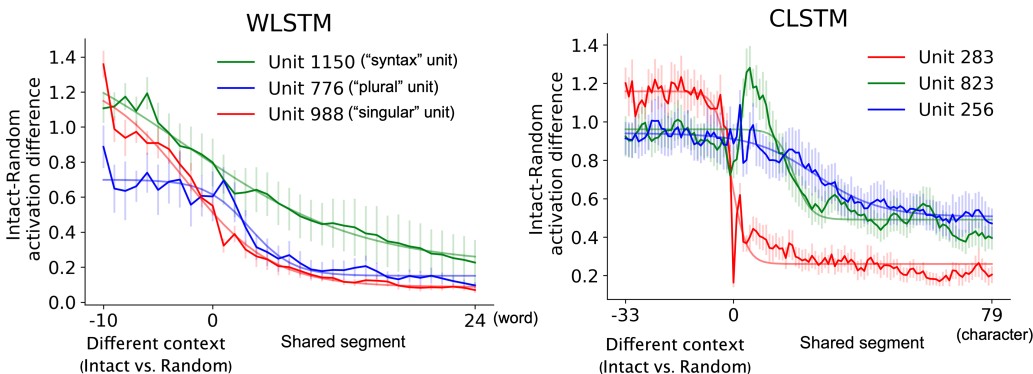

Figure A.1: Example single units activation differences across tokens in word-level LSTM (WL-STM) and character-level LSTM (CLSTM) language models. **A.** Single units activation differences across tokens in the WLSTM, with logistic curve fits overlaid. The error bars indicate 95% confidence interval across trials. The example units include the functional units reported by (Lakretz et al., 2019). **B.** Example units activation differences in CLSTM model. The three units were randomly selected from long-, medium- and short-timescales units.

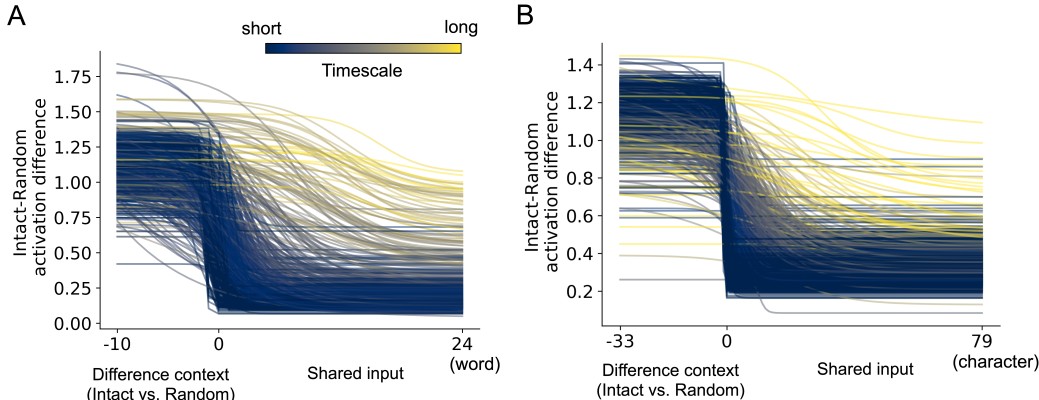

Figure A.2: Logistic fitted curves of activation difference in individual units in word-level and character-level LSTM language model, colored by timescale. **A.** Logistic fitted curves of activation difference over time of all units in word-level LSTM. The color indicates the timescale measured by full-width half-maximum (FWHM) of the curve. **B.** Logistic fitted curves of activation difference over time of all units in character-level LSTM.

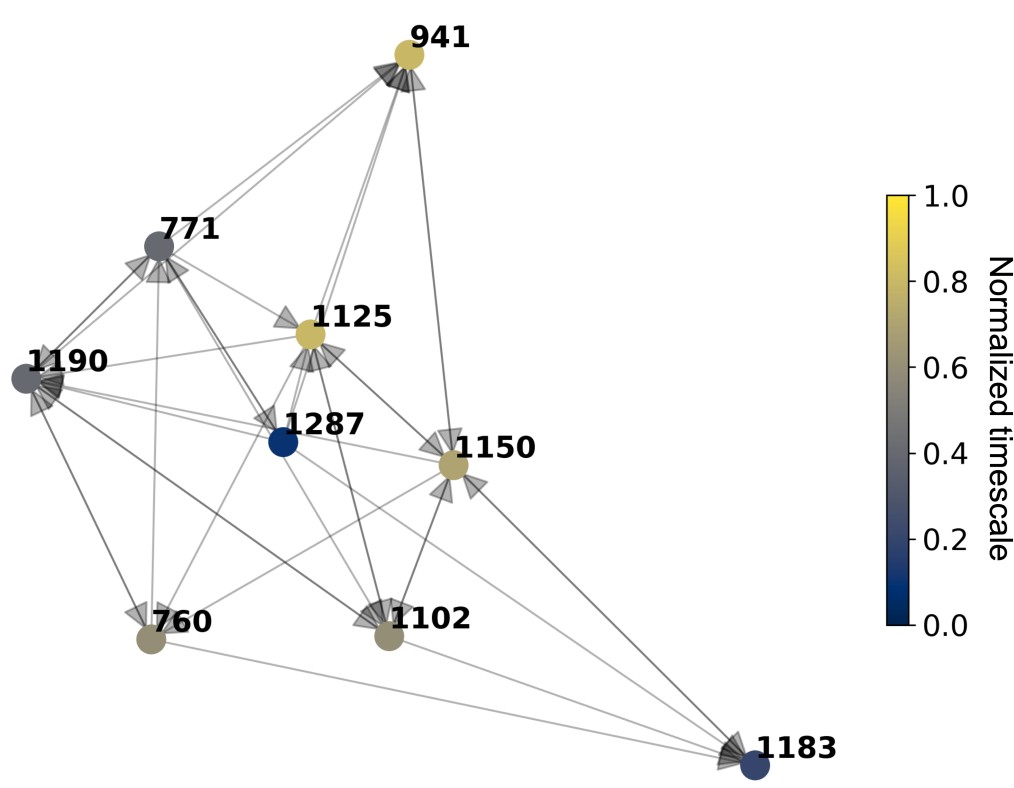

Figure A.3: The core network consists of "controller units" identified in word-level LSTM model.

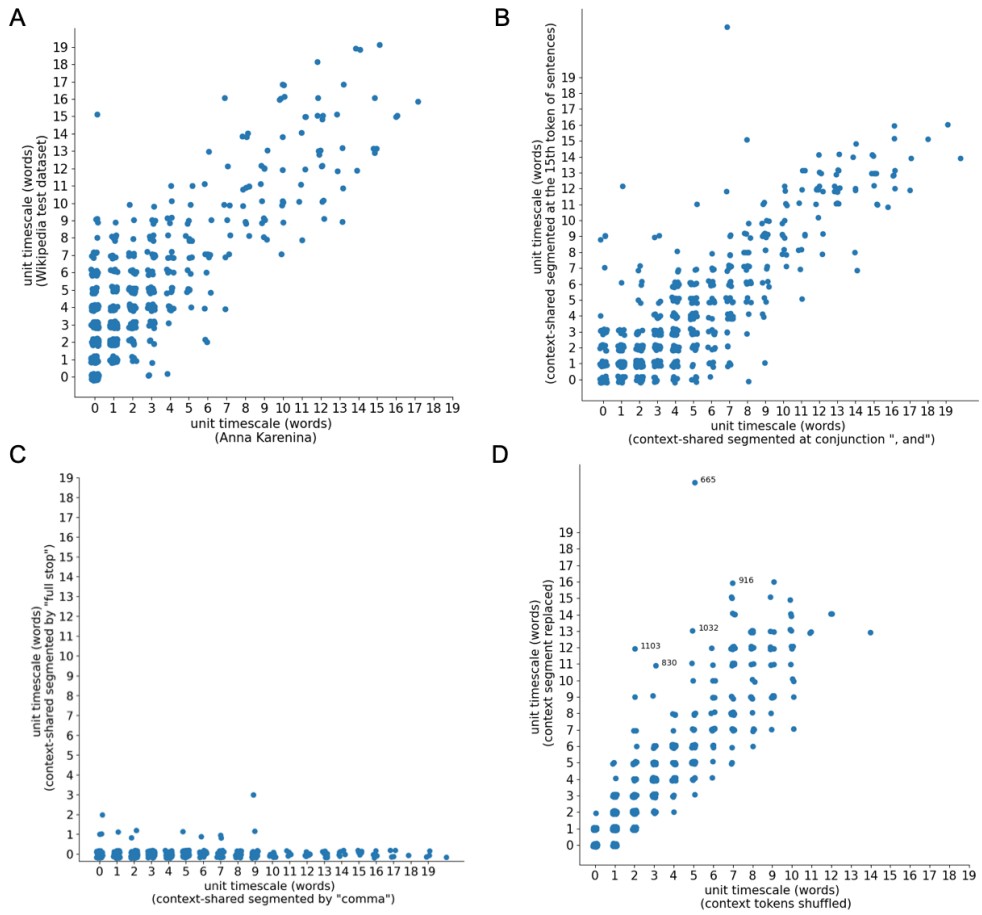

Figure A.4: Mapped timescales across different datasets and context conditions. **A.** Timescales measured using Wikipedia test set from Gulordava et al. (2018) are highly correlated with the timescales measured using *Anna Karenina* ($r$=0.82, p<0.001). **B.** Timescales measured in the middle of a sentence (based on the token number) are highly correlated with timescales measured at the conjunction of a sentence(", and") ($r$=0.83, p<0.001). **C.** Timescales measured between sentences (i.e., using "full stop" to segment context and shared segments), vs. within sentence (i.e., using "comma" to segment context and shared segments). Most of the units showed little context dependence when the segmentation performed at the beginning of a sentence, suggesting a "reset" of context representation in these units. **D.** Timescales measured by replacing the context with syntactically structured segments vs. with the shuffled context tokens. Most of the units showed similar timescales under the two conditions, and several units showed longer timescales (i.e. preserved more context) when the context segments were syntactically structured.

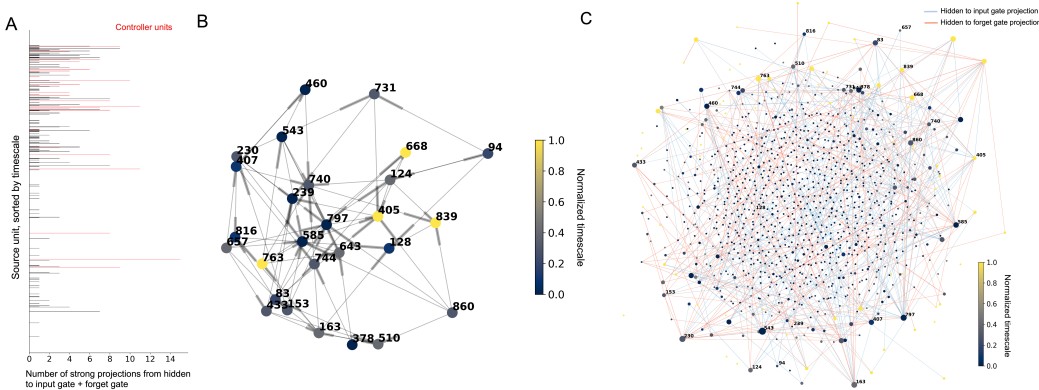

Figure A.5: Timescale and connectivity organization in a character-level LSTM (CLSTM). **A.** Longer-timescale units have overall stronger projections from the hidden units at time $t$ to the forget gates and input gates at time $t + 1$. **B.** The main core network ($k = 4$) formed by the controller units in the CLSTM. **C.** The multidimensional scaling space of the hidden-to-gate connection pattern of all units. The distance between nodes indicates their hidden-to-gate connection similarity; the size of the node indicates the number of strong projections from the node; and the line between two nodes indicates a significant projections between them.

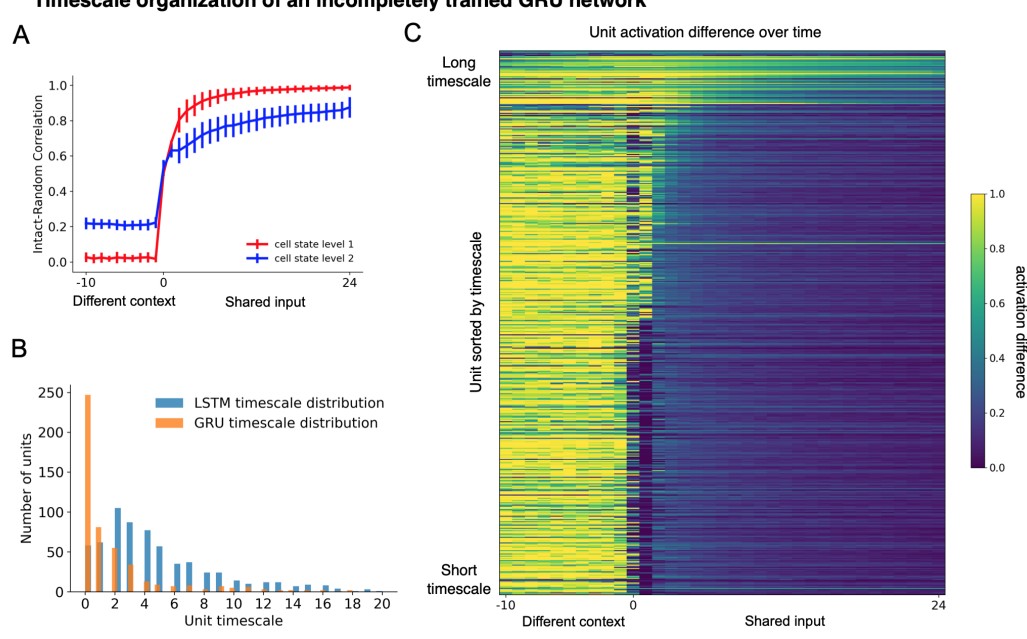

Figure A.6: Mapped timescale organization in an *incompletely trained* GRU language model. **A.** Correlation curves of the GRU hidden-state vectors across the Intact Context condition and Random Context condition as a function of input token. Similar to the word level LSTM, the second layer of the GRU model was more sensitive to prior context than the first layer. **B.** Both GRU and LSTM models showed sparse long-timescale units compared to rich short-timescale units. However, the unit timescale distribution in the incompletely trained GRU is more right-skewed, indicating that the long-timescale units have not emerged due to insufficient training. **C.** Absolute activation difference for each second-layer (incompletely trained) GRU unit over time.

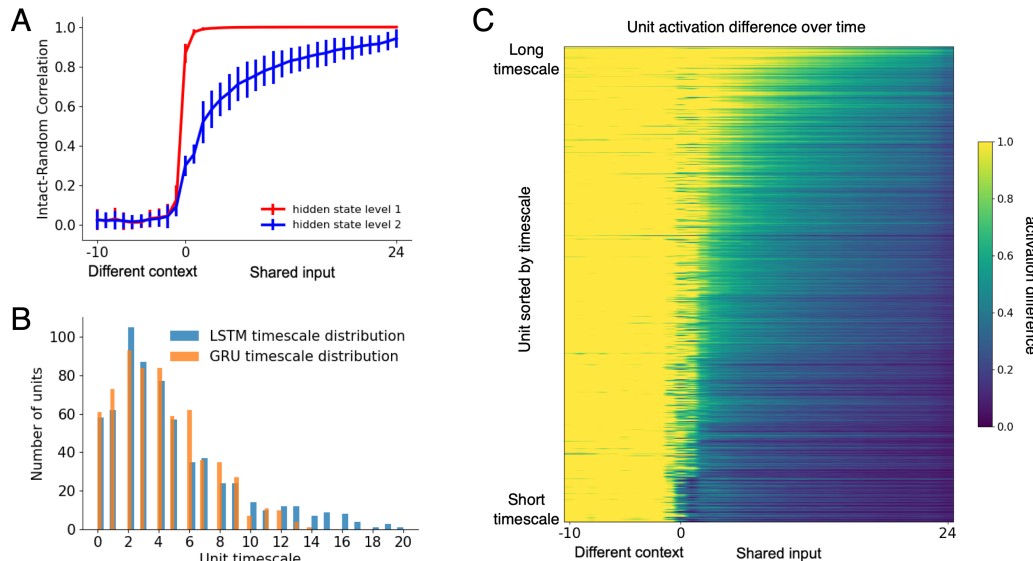

Figure A.7: Mapped timescale organization in a GRU language model. **A.** Correlation curves of the GRU hidden-state vectors across the Intact Context condition and Random Context condition as a function of input token. The second layer of the GRU model was more sensitive to prior context than the first layer. **B.** Distributions of unit timescales are similar in GRU as in LSTM. Both models showed sparse long-timescale units compared to rich short-timescale units. **C.** Absolute activation difference for each second-layer GRU unit over time.

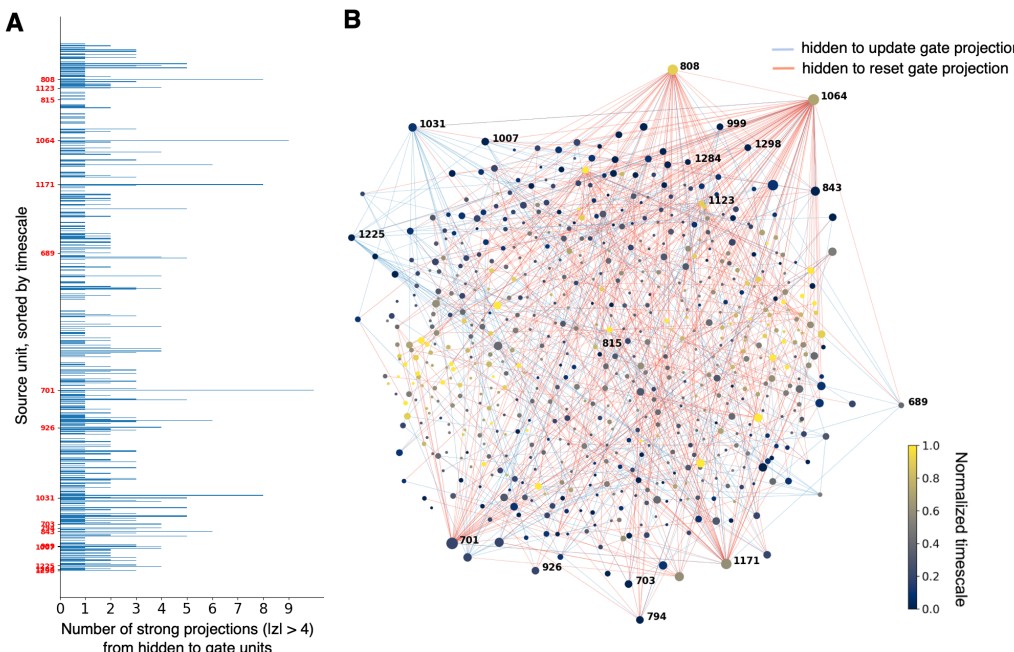

Figure A.8: Timescale-connectivity analyses in a GRU language model. **A.** Different from LSTM, the GRU we analyzed did not show the pattern that units with longer timescales exhibited more strong projections (r=-0.10, p=0.01). **B.** Multi-dimensional scaling (MDS) of connectivity similarity in GRU. The size of each node indicates its degree. An edge between nodes indicates a significant hidden-to-gate projection between them. The "controller units" identified in GRU using k-core analysis (labeled on the graph) tend to locate at the edge of the MDS space, similar to the LSTM. However, we did not observe the "integrator units" in the center of the MDS space. The long-timescale units in the GRU are more distributed in the MDS space compared to those in the LSTM.

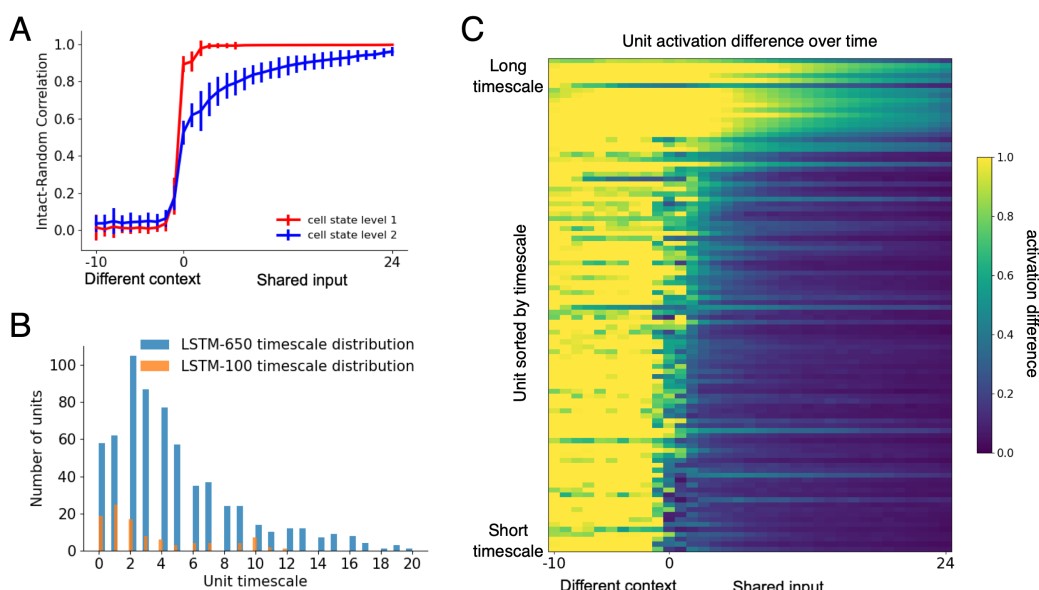

Figure A.9: Mapped timescale organization in an LSTM language model with 100 hidden units per layer (LSTM-100). **A.** Correlation curves of the LSTM cell-state vectors across the Intact Context condition and Random Context condition as a function of input token. Similar to the word-level LSTM reported in the main text, the second layer of the LSTM-100 model was more sensitive to prior context than the first layer. **B.** Distributions of unit timescales in LSTM with 100 units and in LSTM with 650 units. Both models showed sparse long-timescale units compared to rich short-timescale units. **C.** Absolute activation difference over time of LSTM-100 second-layer units sorted by timescale.

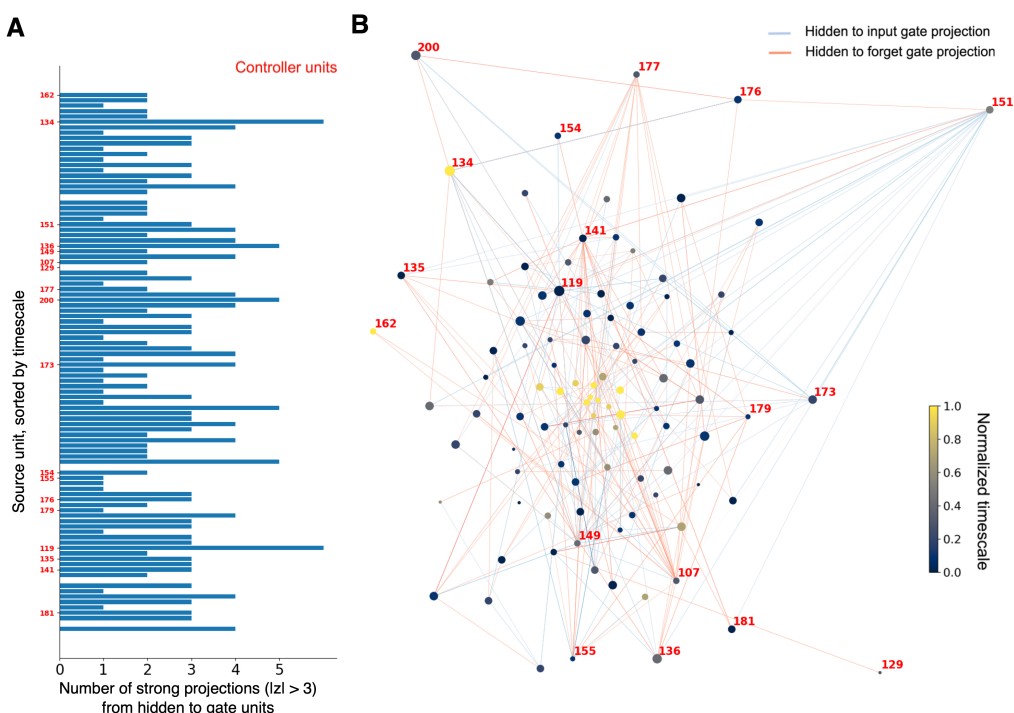

Figure A.10: Timescale-connectivity results in an LSTM language model with 100 hidden units per layer (LSTM-100). **A.** Different from the LSTM with 650 units, LSTM-100 did not show the pattern that units with longer timescales exhibited more strong projections (r=-0.05, p=0.64). **B.** Multi-dimensional scaling (MDS) results in LSTM-100. Similar to LSTM with 650 units, the "controller units" identified in LSTM-100 using k-core analysis (labeled red on the graph) tend to locate at the edge of the MDS space, similar to the LSTM. Furthermore, there are several long-timescale units located in the center of the MDS, analogous to the "integrator units" in the LSTM with 650 units.

