# OpenReview forum: "Mapping the Timescale Organization of Neural Language Models"
_ICLR.cc/2021/Conference — ICLR 2021 Poster_

### Official Review · AnonReviewer3 · 2020-10-19
**Great to see some methods from neuroscience applied to interpretability research for a relevant question, results and setup could be improved**

**Rating:** 3
**Confidence:** 4

**Review:**

_**Update after author response**_: I think this is a very promising paper, and I am really excited about seeing techniques from neuroscience employed to answer questions about neural network models. The authors have further conducted several additional experiments after reviewer comments, which I appreciate. However, my most fundamental concern -- the mismatch between the method and the way that it is validated -- unfortunately still stands, which is why I would encourage the authors to further pursue this line of work, but recommend to reject it for ICLR.

**Summary**

This paper proposes to apply time-scale methods from neuroscience to investigate the timescale organisation in neural language models. More specifically, the authors test the timescale of individual units in a word- and character-level LSTM by comparing the units' activations values on the same sentence, but with different contexts. Using this method, the authors first  show that the higher layers on average have longer timescales. They then, for all units, they fit a logistic function to the "recovery" curves and use the half-times of this curves as an indication of the time scale of these units. They test the syntax unit and two long-distance units found by Lakretz et al and show that the number units have similar time-scales, while the syntax unit have a longer time scale. Lastly, the authors analyse the connectivity between the longer time scale units and find that the units with longer processing timescales make a larger number of strong projections. Within these units, the authors identify two sets of units in the word-level LSTM: "controller units", that play a role in how the connectivity of the network is updated, and "integrator units", that instead integrate information.

**Strong points**
- Neuroscience has long been asking questions about the brain that are very similar to the questions we now ask about neural networks, cross-pollination between these fields is extremely important, and this paper contributes to this
- Aside from the main technique, the paper introduces some interesting and useful methods, such as projectivity analysis and k-core analysis. I think these methods can be useful for other researchers as well
- Time scale analysis of LSTMs is a very relevant and interesting topic, that deserves more attention than it is currently getting

*Concerns*
- My main concern is that there seems to be a mismatch between the "language time scales" on which the authors operate: their experiment is designed to investigate the impact of extra-sentential context, but the Lakretz et al results they keep coming back to concern syntactic phenomena that are only relevant *within* a sentence, which is a different scale. In other words, the units found by the authors of this paper are long-distance when it comes to integrating context, but the syntax and number units found by Lakretz et al are not really related to that: they model relationships *within* sentences. Theoretically speaking, they should be reset at the beginning of every new sentence and they should thus be completely independent from the content. That the authors find this to be untrue is interesting, but inconsistent with what Lakretz et al describe these unit do. Since this is not addressed at all in the paper, it makes the results in general a bit difficult to interpret. _**Update after author response**: In their response the authors clarified that the they have only analysed single sentences, where two distinct subsentences are combined with a conjunction. This, unfortunately, does not make a difference for the argument: whether two sentences are split by a full stop or instead concatenated with "and" does not make any difference for the argument above, since the subject-verb agreement relationships that the units the authors look at model do not cross these boundaries either. Furthermore, in their response the authors state that the find that the context representations of units was 'reset' at sentence boundaries, as I asked before. I appreciate that the authors did these additional experiments, but I find the result somewhat worrisome: since the units they are looking at are syntactic units that encode number across long distance subject verb relationships, they should be reset both when a new sentence starts, as well as when a new conjunct with a new relationship starts. In terms of SV relationships, there should be no difference between "The boy kicked the ball and the girl caught it" and "The boy kicked the ball. The girl caught it." That the authors do find a difference points to a potential flaw in methodology._

- Relatedly, the authors say that  their result that the syntax unit is a long distance unit, while the number units are not. This is not consistent with what they say in the related work of the section, but also not with the results reported by Lakretz et al, who hypothesise that the syntax units represent the depth of the syntactic dependency. This is something that changes with every new incoming word, whereas the number units are the ones that have to keep their activation constant across time.

- While, as I said before, I think it is great that the authors try to use methods from neuroscience into the field, I do think that in this case  the main method they propose is only very marginally different from earlier work (in particular Khandelwal et al. Perhaps it would make more sense to put a bit more stress on the rest of the methods as well (btw, also Lakretz et al do connectivity analysis).
- The results are a bit underexplained, and understanding them requires many back and forths to the appendix. I would have appreciated a bit more motivated interpretation of several aspects. For instance: why is there such a large difference in activation differences in different units in the "pre-shared segment" part, and is this related to the half-time (it seems so from the plots)? What is the difference between character and word-level models in terms of expectations (we'd expect there to be an additional level of time-hierarchy, perhaps?) How do assessing activation differences and correlations differ in terms of conclusions? These things should, in my opinion, all be worked out a bit better.
- Lastly, there are a few unsupported claims, the most important of which that their method recovers the previously discovered units of Lakretz et al, while (as far as I understand), they actually only *use* their method to analyse those neurons, but did not find them independently. (for other suggestions and comments, see below).

To summarise, while I think the idea is very nice and definitely worth working out further, I do think that some work is needed to make this a publishable paper.


*Suggestions/comments for authors*
_Typographic_:
- If you use quotes in latex, you should use different ones for left (`) and right ('), for them to appear correctly (check for instance line three in the introduction)
- To prevent additional spaces after abbreviations like e.g. and i.e., put a backslash: "e.g.\ "
- Lerner et al --> put all references within parenthesis
- Introduction switches from present tense to paste tense in the last paragraph
- "we measure the time-taken for the effect of this prior context to ”decay” (see Methods)" --> I don't really understand what this means, you measure how long it takes for these changes to not be measurable anymore?
- Try to avoid double parethesis with abbreviations, e.g.: (WLSTM Gulordava et al. (2018)) should be: (WLSTM, Gulordava et al; 2018). You can do this with \citep[text before][text after]{citation}.
- "has an 650-dimensional" --> "has a 650-dimensional"
- "without fine-tuning to the novel" --> I first thought this sentence was unfinished until I read back and realised that "the novel" is your corpus. This is a bit confusing perhaps you could rephrase.
- "how the cell state activation differ" --> "how the cell state activations differ"
- "we will see that the activation difference drop quickly' --> drops quickly / see the activation difference drop quickly
- There are several references that were published at ACL* conferences that are listed as arxiv papers in the reference list (Lakretz et al, Gulordava et al, Khandelwal et al)

_Content_
- I would say that the conclusion that "Overall, prior works suggests that a small subset of units track long-range dependencies" is rather overstated: Lakretz et al found that the units representing long distance number information were sparse, but this does not imply that long range information in general is represented sparsely. Their method also focusses quite exclusively on finding sparsely distributed properties, as more distributed properties cannot be found with ablation. Furthermore, this is just one study, focusing on one syntactic aspect. I would suggest to rephrase this a bit.
- Lakretz at all actually identified several syntax units, but only one of them was interpretable.
- I find it a bit confusing that in 3.2, second paragraph, you first talk about comparing cell state activation, then say that you compare hidden state activations and then talk again about the cell state activation

- Figure 1 C & D: I don't think these figures add much to the paper, for the following reasons i) They show only individual units and no average, making it difficult to interpret the values ii) while, as pointed out in 5.1, the *rate* of decay is the most important, the cut-off point is not indicated in the figure, which puts a stress on irrelevant aspects: the actual difference between the two lines.
- I would appreciate to have Figure A.1 in the main text, it is important for the story.

---

> ### Author Response · Authors · 2020-11-20
> **Official response to Reviewer 3**
>
> We thank the reviewer for acknowledging the potential contribution of this paper, including the importance of transferring methods from brain science to understand neural network AI models, and the importance of analyzing timescales in neural language models.
>
> Here are the point-by-point responses to the concerns raised by the reviewer:
>
> - *My main concern is that there seems to be a mismatch between the "language time scales" on which the authors operate: their experiment is designed to investigate the impact of extra-sentential context, but the Lakretz et al results they keep coming back to concern syntactic phenomena that are only relevant within a sentence, which is a different scale. In other words, the units found by the authors of this paper are long-distance when it comes to integrating context, but the syntax and number units found by Lakretz et al are not really related to that: they model relationships within sentences. Theoretically speaking, they should be reset at the beginning of every new sentence and they should thus be completely independent from the content. That the authors find this to be untrue is interesting, but inconsistent with what Lakretz et al describe these unit do. Since this is not addressed at all in the paper, it makes the results in general a bit difficult to interpret.*
>
> We apologize for the confusion here. Crucially, our work only examines phenomena within an individual sentence, just as in the Lakretz study. The reviewer is correct that context representations are ‘reset’ at sentence boundaries in the Gulordava et al. model (see below, where we confirm this in our own data). For this reason in an early draft of our manuscript, we had only analyzed single sentences which combined two distinct sub-sentences in the following way, e.g.:
>
> “The boy kicked the ball, and the girl caught it.”
>
> Since the segment before the conjunction “, and” can be read as a self-contained sentence, our original paper contained text such as : “the preceding sentence differed across the two conditions” (in Section 3.2). However, to be clear, these preceding segments were always part of the same sentence. Thus, in all of the data reported in our paper, we only examined context effects within a single sentence. We apologize again for the confusion. For improved clarity, we have now revised the manuscript so that the word “segment” is used consistently throughout to refer to the prior context segment and to the shared input segment.
>
> Consistent with the reviewer’s statement, we did find that the context representation of units was “reset” at sentence boundaries. To demonstrate this, we examined the timescales of each unit when the context segment and shared input segment were separated by a “full stop” symbol, which signals the end of a sentence. (Please refer to the result figure here: https://anonymous.4open.science/repository/ef3696c5-e97e-4bd8-b12a-94795a038b8c/fullstop_reset.png). When the first and second segments were separated “full stop” symbol, we found that the timescales inferred for most of the units became extremely short, compared to when the first and second segments were separated with a “comma” symbol as in our original paper. This “reset” phenomenon at the beginning of every new sentence is consistent with what the reviewer predicts and with the results of Lakretz et al.

---

> > ### Author Response · Authors · 2020-11-20
> > **Official response to Reviewer 3 (cont.)**
> >
> > - *Relatedly, the authors say that their result that the syntax unit is a long distance unit, while the number units are not. This is not consistent with what they say in the related work of the section, but also not with the results reported by Lakretz et al, who hypothesise that the syntax units represent the depth of the syntactic dependency. This is something that changes with every new incoming word, whereas the number units are the ones that have to keep their activation constant across time.*
> >
> > Thank you for raising this important issue. Indeed, we can distinguish two ways of measuring “distance” between the current input and its prior context. On the one hand, distance can be measured with reference to an abstract language structure (a “functional” type of distance). On the other hand, distance can be measured with the a simple count of the number of “tokens” (an “implementation” type distance). We feel that both notions of distance are valuable for understanding how LSTMs represent and process linguistic information.
> >
> > We agree with the reviewer that if the syntax unit is tracking the depth of syntactic hierarchy, as Lakretz et al. hypothesized, then its timescale of context dependence should be flexible and? based on the syntactic structure of a sentence.
> >
> > Conversely, one could also imagine units whose maintenance of information varies explicitly as a function of token distance, because, after all, recurrent neural networks face the challenge that prior context must be passed forward timestep-to-timestep. The need to preserve context from token to token is why the vanishing and exploding gradients problem become so salient for long-range dependencies.
> >
> > In this paper, we demonstrated a model-free method for measuring the “average timescales” of individual units to aid in understanding the functional organization of the system. While other methods can certainly be used to measure timescales under more carefully controlled contexts, or to test specific functional models, we feel that the method we used has the following advantages: (i) it can be applied to a wide range of architectures, corpora and languages in a manner that is agnostic to the functional structure of the language; (ii) it maintains a connection to the implementation-level constraints faced by the LSTM by measuring the token-level distance.
> >
> > The syntax unit and number units are both medium-to-long timescale units relative to other units (Figure 2A) when the timescale is measured as the average number of tokens of prior context that affect the current response. Although it is beyond the scope of this project, a fascinating question for future work would be to examine the relationship between the timescale map measured in “tokens” as presented here, and the timescales that would be derived based on more functional metrics such as syntactic distance.

---

> > > ### Author Response · Authors · 2020-11-20
> > > **Official response to Reviewer 3 (cont.)**
> > >
> > > - *While, as I said before, I think it is great that the authors try to use methods from neuroscience into the field, I do think that in this case the main method they propose is only very marginally different from earlier work (in particular Khandelwal et al. Perhaps it would make more sense to put a bit more stress on the rest of the methods as well (btw, also Lakretz et al do connectivity analysis).*
> > >
> > >
> > > Although we were certainly inspired by the work of Khandelwal et al. and by Lakretz et al., our work has goals and findings that are clearly distinct from both.
> > >
> > > Khandelwal et al. used a model-agnostic method to measure how much context is used by the LSTM as a whole; they measured how the overall model performance (measured by loss and perplexity) was affected when the given context was limited or scrambled. Khandelwal did not seek to understand the variation of timescales within different components of the model architecture, or the information flow between the different components.
> > >
> > > Lakretz et al. were interested in the contextual representations of individual units within the LSTM. Their approach was not model-agnostic; instead, their context measurements related to specific functions (e.g. tracking of number). Therefore, they did not employ a context-scrambling procedure, and did not set out to map the overall profile of contextual processing across the model architecture.
> > >
> > > In the present paper, we set out to map how much context is encoded by individual units in LSTMs, and to understand the flow of information between units of shorter and longer timescales. Therefore, we proposed a method for mapping the timescales of context dependence in individual units. Further, we explored the relationship between the mapped timescale of each unit and its role in the LSTM network structure. Neither Khandelwal et al. nor Lakretz et al. examined the large-scale functional architecture of the network. Thus, for example, these prior studies did not characterize how many nodes in the LSTM engaged in relatively context-free processing, nor how many different nodes tracked long-range context dependencies. We were certainly inspired by Khandelwal et al. and by Lakretz et al. (as well as by recent developments in neuroscience); at the same time, we hope it is clear that our goals and findings are distinct.
> > >
> > > We have revised the Introduction to make these distinctions clearer.
> > >
> > > - *The results are a bit underexplained, and understanding them requires many back and forths to the appendix. I would have appreciated a bit more motivated interpretation of several aspects. For instance: why is there such a large difference in activation differences in different units in the "pre-shared segment" part, and is this related to the half-time (it seems so from the plots)?*
> > >
> > > Thank you for the questions. As another reviewer also suggests, we will reorganize the Appendix to make it easier to understand. We will also add some of these points below into the manuscript. Below are our responses to some of the specific questions.
> > >
> > > The variation across units in “activation differences” when processing the different context segments could be driven by two sources:
> > > (1) individual units have different functions in language processing; and
> > > (2) the linguistic content of the “pre-shared segment” caused units to have different activations accordingly.
> > >
> > > The variation across units in the height of the asymptote (“pre-shared segment”) is most likely due to the first factor. If a unit tracks a linguistic feature that varies very little across most sentences, then its activation patterns will be quite similar across many different sentences.
> > >
> > > We did not observe a relationship between the inferred processing timescales (the half-time) and the activation difference during the context period (the asymptote during the context segment). We computed the correlation between these two variables across units. (Please refer to the figure here: https://anonymous.4open.science/repository/ef3696c5-e97e-4bd8-b12a-94795a038b8c/time_asym_corr.png).
> > >
> > > We conducted this analysis in two datasets: the dataset used in our early draft manuscript (novel Anna Karenina), and the test dataset used in Gulordava et al. derived from the Wikipedia corpus used to train this LSTM model. As the figure shows, for both datasets, there is very little relationship between the activation difference in the context period and the timescale inferred for each unit. That said, in future work, we will be investigating the reasons for the differences in the heights of the asymptotes of the logistic fits.
> > >
> > > It also bears mentioning that there does appear to be a relationship between the processing timescale and the asymptote on the right-hand side of the curve (i.e. the magnitude of activation difference after a unit has already processed 10 or more tokens of shared context), but this is expected, since nodes with longer processing timescales, should show a larger difference across contexts.

---

> > > > ### Author Response · Authors · 2020-11-20
> > > > **Official response to Reviewer 3 (cont.)**
> > > >
> > > > - *What is the difference between character and word-level models in terms of expectations (we'd expect there to be an additional level of time-hierarchy, perhaps?)*
> > > >
> > > > In terms of expectations between word-level and character-level models, under the hypothesis that character-level models can achieve similar performance as word-level models (as reported in Hahn et al. 2019), one should expect to see character-level models have units with longer timescales (measured at the scale of “character” tokens, not “word” tokens) than word-level models. This would be necessary to allow character level models to integrate not only word-level but also sentence-level information. Indeed, we discovered units with timescales up to 50 tokens in the character-level model, which are much longer than the longest timescale of any unit in word-level model (~20 tokens). Still, this is just a rough comparison since the word-level and character-level model we evaluated in the study were trained with different architecture, and the training corpora (Wikipedia dataset) were preprocessed differently as well.
> > > >
> > > > In Figure 2, we do not observe any obvious two-scale structure in the character-level LSTM, with a subset of nodes integrating characters at the word scale, and a separate set of characters integrating word-level information into larger structures. However, this is an interesting proposal which could guide future work – for example, the prior context could be manipulated at a much finer grain in the LSTM, which could reveal a gradation of units that focus on within-word contextual integration. Since we focused on Layer 2 of the LSTM models, it may be more fruitful to conduct such an intra-word analysis in Layer 1 of the model, which has a shorter timescale overall.
> > > >
> > > > - *How do assessing activation differences and correlations differ in terms of conclusions?*
> > > >
> > > > All of our analyses of individual unit timescales (i.e. via activation differences) were performed within Layer 2 of the LSTM model. All of our analyses of correlation were applied at the level of an entire LSTM layer (e.g. layer 1 or layer 2). Thus, the correlation-based analyses and activation-based analyses were not applied to address the same questions in this paper.
> > > >
> > > > In general, one would expect that a higher cross-context correlation would correspond to lower activation difference across contexts, indicating less representational change.  Thus, within layer 2, if we manipulated the network so as to reduce the timescales inferred from the activation patterns of individual units, we would also expect this effect to show up in the aggregate and would reduce the timescale inferred from measuring the state of the entire network-layer using the correlation approach.
> > > >
> > > > We hope that the text above clarifies the relationship. We have adjusted the Methods text to emphasize that the unit-level activation difference analyses were only applied in Layer 2.
> > > >
> > > >
> > > > - *Lastly, there are a few unsupported claims, the most important of which that their method recovers the previously discovered units of Lakretz et al, while (as far as I understand), they actually only use their method to analyse those neurons, but did not find them independently.*
> > > >
> > > >
> > > > Thank you for raising this point. We have changed the wording in the abstract, so that, instead of saying that we “recovered” the units, we say that our method was “validated against” these units, to make clear that these units were discovered by a different approach, and served as a reference.
> > > >
> > > > The main focus of the paper is to propose a model-agnostic approach to measure timescales of all units in the network. From this perspective, it was valuable to be able to validate our method by showing that the syntax unit and number units (identified by Lakretz et al) exhibited relatively long timescales. Our goal in this paper is to emphasize the overall timescale organization of the network, and the existence of other long-timescale units.
> > > >
> > > > If possible, we would appreciate if the reviewer could point out any other specific sentences that contain unsupported claims, so that we can address them carefully.
> > > >
> > > > - *Suggestions/comments for authors Typographic:*
> > > >
> > > > Thank you for the suggestions regarding the format and typos in the manuscript! We will revise the manuscript carefully based on the suggestions and comments.

---

> > > > > ### Author Response · Authors · 2020-11-20
> > > > > **Official response to Reviewer 3 (cont.)**
> > > > >
> > > > > - *Content*
> > > > > - *I would say that the conclusion that "Overall, prior works suggests that a small subset of units track long-range dependencies" is rather overstated: Lakretz et al found that the units representing long distance number information were sparse, but this does not imply that long range information in general is represented sparsely. Their method also focusses quite exclusively on finding sparsely distributed properties, as more distributed properties cannot be found with ablation. Furthermore, this is just one study, focusing on one syntactic aspect. I would suggest to rephrase this a bit.*
> > > > >
> > > > > We agree with the reviewer. We will rephrase this sentence in our revised manuscript and will emphasize that while Lakretz et al. showed that number and syntax information were sparse, it remained unknown whether long-range context representations were sparse in general.
> > > > >
> > > > > - *Lakretz et al actually identified several syntax units, but only one of them was interpretable.*
> > > > >
> > > > > Thank you for mentioning this; we will add this information into the manuscript.
> > > > >
> > > > > - *I find it a bit confusing that in 3.2, second paragraph, you first talk about comparing cell state activation, then say that you compare hidden state activations and then talk again about the cell state activation*
> > > > >
> > > > > Thank you for pointing this out. We will revise the manuscript to make it consistent.
> > > > >
> > > > > - *Figure 1 C & D: I don't think these figures add much to the paper, for the following reasons i) They show only individual units and no average, making it difficult to interpret the values ii) while, as pointed out in 5.1, the rate of decay is the most important, the cut-off point is not indicated in the figure, which puts a stress on irrelevant aspects: the actual difference between the two lines.*
> > > > >
> > > > > Thank you for the suggestion!  We will rearrange the figures in our revised manuscript.
> > > > >
> > > > > - *I would appreciate to have Figure A.1 in the main text, it is important for the story.*
> > > > >
> > > > > Thank you for pointing this out. We will rearrange the figures in the main text and appendix accordingly.

---

### Official Review · AnonReviewer1 · 2020-10-27
**A promising approach to explore the emergent structure of LSTM models, which needs further development to shed light on emergent function**

**Rating:** 6
**Confidence:** 4

**Review:**

This paper explores the application of innovative methods to track the flow of linguistic information in LSTM language models. In particular, the overarching question is how contextual information might be encoded in the network at the level of single units, and how context disruption might alter the LSTM dynamics and thus impact its predictive ability.
The paper is clear and it tackles an interesting question. The approach is well motivated, and the authors give a brief survey of the most recent applications of this kind of methodology in linguistics and cognitive neuroscience studies.
The methodology is generally appropriate, though some details and parameters (e.g., numerical thresholds) seem to be chosen arbitrarily. Also, the analysis could be improved by applying statistical testing in order to better quantify the strength of the observed effects.
Overall, I think this is a nice paper, though it might be especially relevant to the linguistics community rather than to the ICLR community. Moreover, I think that further analyses are required in order to better clarify some important aspects. In particular, I think that ablation studies should be performed in order to better identify the functional role of the “controller” and “integrator” units, whose actual functional role remains a bit speculative (and mostly based on structural / connectivity information). It would also strengthen the paper to have some more controlled simulations, where the contextual information is defined according to specific linguistic constraints, in order to better characterize what the target units are actually encoding. Indeed, as also noted by the authors almost “all the long timescale units are of unknown function”. Finally, I think that it would be important to establish whether these findings are generally applicable to LSTM models, regardless of the specific architecture under investigation (e.g., What happens if we force the LSTM to rely on fewer units? Does the hierarchical organization of the context improve by adding more layers?).

Other comments:
- Why did the author choose to test the model on a different corpus (Anna Karenina novel) rather than considering a test set from the same corpus from which the training set was derived? The Tolstoy book might have a quite different linguistic structure from that of the corpora used to train the LSTMs.
- It might be informative to also include a third condition in-between “Intact” and “Random” context, where the same context words are maintained with scrambled order. This would allow to better understand the role of individual words in shaping context representation and activating the LSTM units.
- In Fig. 1D, it is interesting to note that the Unit 823 (green line) actually exhibits a sharp increase in difference after the shared segment starts. Do the authors have a possible explanation for this kind of phenomena? Was it observed systematically in other units?
- In relation to the results shown in Fig. 3A, I did not understand how the thresholds and parameters for the k-core analysis were chosen.
- Pg. 3: there is a typo regarding the size of the output layer (5,0000)
- In Fig. A1, error bars would help in better understanding the actual difference between the curves.
- In order to improve reproducibility, it would be very helpful to share the source code used for these analyses.

---

> ### Author Response · Authors · 2020-11-20
> **Official response to Reviewer 1**
>
> Thank you for your positive comments! Please see our responses to the points you raised below:
>
> - *I think that further analyses are required in order to better clarify some important aspects. In particular, I think that ablation studies should be performed in order to better identify the functional role of the “controller” and “integrator” units, whose actual functional role remains a bit speculative (and mostly based on structural / connectivity information). It would also strengthen the paper to have some more controlled simulations, where the contextual information is defined according to specific linguistic constraints, in order to better characterize what the target units are actually encoding. Indeed, as also noted by the authors almost “all the long timescale units are of unknown function”.*
>
> Thank you for the positive comments and valuable suggestions. We agree that it is important to further investigate the functional roles of the units we identified using the current analyses. Each putative “controller unit” may serve a different functions due to its distinctive positions in the MDS space (Figure 3D). It will requiring targeted linguistic experiments to probe each functions. Nonetheless, we performed a preliminary group ablation analysis to look at how ablating the controller units influences model performance, relative to the intact original model, and relative to the ablation of a random set of units.
>
> We evaluated model performance by examining the difference of probabilities assigned to the target words in the ablated model and in the intact model, and comparing the conditions when controller units (N=10)/integrator units (N=5) were ablated vs. same number of random units from layer 2were ablated. We used the test corpus used by Gulordava et al. and measured the average performance of each model across 100 text-batches, randomly sampled from the Wikipedia test dataset. Each text-batch was composed of 1000 tokens start from the beginning of a sentence.
>
> We found that ablating controller units reduced the probabilities the model assigned to the target words, more so than ablating random units (controller vs. random across 100 text batches: Cohen’s d = 4.96, t=-35.12, p<0.001). Ablating integrator units did not show significant difference compared to ablating random units (Cohen’s d = 0.8, t=1.77, p=0.09). It may be that the integrator units mostly influence the model performance on predicting tokens in cases where long-range information is especially relevant (e.g. in the later portions of long clauses). Overall, these abalation results support the non-trivial functional role for the controller units, which are only 10 amongst 650 units in total. (Please refer to the figures here: https://anonymous.4open.science/repository/ef3696c5-e97e-4bd8-b12a-94795a038b8c/ablation_analysis.png)
>
> However, as we mentioned, this only provides a rough sense of the importance of these units compared to other units in the network. Also, as Reviewer 4 commented, ablation analysis has its limitation, and one should be careful interpreting the results (https://doi.org/10.1007/s42113-020-00081-z). Analyses targeted on specific linguistic properties will be required to further understand the exact functional role of each unit in the controller network, in the vein of Lakretz et al. (2019), and we hope that the timescale and network mapping methods we have introduced here can help to target future investigations of this kind.
>
> - *Finally, I think that it would be important to establish whether these findings are generally applicable to LSTM models, regardless of the specific architecture under investigation (e.g., What happens if we force the LSTM to rely on fewer units? Does the hierarchical organization of the context improve by adding more layers?).*
>
> Reviewer 2 has also raised the concern of whether the results are applicable to other architectures. To address this concern, we trained a GRU language model, mapped the timescales of each layer, mapped the timescales of individual units, and explored the relationship between timescale and connectivity patterns. We found similarities and differences regarding the results between GRU and LSTM; the distribution of timescales across nodes was similar across the GRU model and the LSTM. The GRU did not show the same timescale-connectivity relationship as the LSTM. These findings remain preliminary because of the limited time available for training the GRU model. For further details, please refer to our response to the third point raised by Reviewer 2.

---

> > ### Author Response · Authors · 2020-11-20
> > **Official response to Reviewer 1 (cont.)**
> >
> > - *Other comments: Why did the author choose to test the model on a different corpus (Anna Karenina novel) rather than considering a test set from the same corpus from which the training set was derived? The Tolstoy book might have a quite different linguistic structure from that of the corpora used to train the LSTMs.*
> >
> > We agree with the reviewer’s point that the two corpora have different linguistic structure. We have re-analyzed the timescales using the test set derived from the Wikipedia training corpora that the model was trained on. We have also increased the sample size of test sentences to 500. We found the timescales measurements were highly correlated across the two corpora (r = 0.82), indicating that our results are not specific to the particular corpus we originally tested. We will revise the manuscript to add our new analyses using this dataset.
> >
> > - *It might be informative to also include a third condition in-between “Intact” and “Random” context, where the same context words are maintained with scrambled order. This would allow to better understand the role of individual words in shaping context representation and activating the LSTM units.*
> >
> > Thank you for the excellent suggestion. We agree that this analysis would be useful to understand the how the context structure shapes the representation of each units. We conducted an analysis similar to that of Khandelwal et al. to map units’ timescales using the shuffled context vs. intact context, and then compared the results with our original analysis in which we replaced the prior context. (Please refer to the figure here: https://anonymous.4open.science/repository/ef3696c5-e97e-4bd8-b12a-94795a038b8c/timescale_corr_middle_shuffle.png).
> >
> > We observed that most long timescale units (timescale > 10 words) showed longer timescales when the context was replaced compared to when it was shuffled. One unit (Unit 665) showed a larger effect than all others, with its timescale estimate decreasing from more than 20 words down to about 6 word for the shuffled context. This phenomenon indicates that many of the long-timescale units are preserving context in a way that depends on the presence of a coherent structure (e.g. grammaticality of the prior context) and that there is a very small subset of long-timescale units that almost entirely resets  their context sensitivity for shuffled text. We did not observe any units which showed longer timescales for shuffled text than for replaced text.
> >
> > All in all, this analysis indicates that for most units the individual words do indeed play a large role in shaping the context representation (because overall the timescale patterns are highly correlated across the shuffling method and the replacement method). At the same time, for the units with the longest processing timescales, the context representations were preserved much longer when that prior context is composed of coherently structured language.
> >
> > We will revise the manuscript to include these new analyses.
> >
> > - *In Fig. 1D, it is interesting to note that the Unit 823 (green line) actually exhibits a sharp increase in difference after the shared segment starts. Do the authors have a possible explanation for this kind of phenomena? Was it observed systematically in other units?*
> >
> > Thank you for raising the question. As shown in Figure 2B, some units in CLSTM seem to exhibit such phenomenon at the beginning of the shared segments (i.e., starting at “, and” in our analyses). We speculate that this might be due to the units’ sensitivity to “start of phrase” / “start of clause” / “syntactic head” which would be revealed based on our method to segment context and shared input segments in the current paper. We are checking this by using different segmentation method (Please see our reply to the first point raised by Reviewer 2) and seeing if this phenomenon in Unit 823 is preserved.
> >
> > - *In relation to the results shown in Fig. 3A, I did not understand how the thresholds and parameters for the k-core analysis were chosen.*
> >
> > Thank you for raising this. We will revise the manuscript to better explain the settings of these analyses. The threshold chosen here does not really change the results shown in Fig. 3A that units with longer timescales tend to have more strong projections: Using |z| > 3 as threshold we obtained corr(timescale, projections) = 0.30, p<0.001; using |z| > 4 we obtained corr(timescale, projections) = 0.35, p<0.001; using |z| > 5 we obtained  corr(timescale, projections) = 0.29, p<0.001. For the k-core analysis, we chose top n weight values form the weight matrices as edges to construct the network, where n is the number of identified strong projections.

---

> > > ### Author Response · Authors · 2020-11-20
> > > **Official response to Reviewer 1 (cont.)**
> > >
> > > - *Pg. 3: there is a typo regarding the size of the output layer (5,0000)*
> > >
> > > Thank you for the correction. We will revise the manuscript accordingly.
> > >
> > >
> > > - *In Fig. A1, error bars would help in better understanding the actual difference between the curves.*
> > >
> > > Thank you for the suggestion. We will add error bars onto the hidden state correlation in the revised manuscript.
> > >
> > >
> > > - *In order to improve reproducibility, it would be very helpful to share the source code used for these analyses.*
> > >
> > > We agree with the reviewer that code sharing is important. We did not do this in the early version of manuscript due to time limit. We will add an anonymous link for sharing the code used for this study in the revised manuscript.

---

> > > > ### Comment · AnonReviewer1 · 2020-11-23
> > > > **The authors put efforts in revising and extending their analyses, but some points are still unclear**
> > > >
> > > > I appreciate the efforts made by the authors to address most of the issues raised during the review process.
> > > >
> > > > I think that the additional analyses (e.g., the ablation study and the simulations with “controlled” random context) shed some light on the computational role of some unit types, but further (maybe future) investigations are required to more fully characterise the functional role of the integrator and controller units.
> > > >
> > > > The comparison with the GRU model is useful, though somewhat limited given that that model was trained sub-optimally. Moreover, I would have liked to see some explorations about few critical hyperparameters of the LSTM (most importantly, number of hidden units) in order to better evaluate the robustness of the results.

---

> > > > > ### Author Response · Authors · 2020-11-24
> > > > > **Thank you for your feedback!**
> > > > >
> > > > > We have some updated results for the ablation analyses which we put in the revised manuscript (Figure 4C, Appendix A.5). In brief, we found that while controller units affected the overall probabilities assigned to the target words, integrator units played a more important role in assigning probabilities to the words in the later part of the sentences. These results further distinguish the functions of these two sets of long-timescale units. Still, we agree with the reviewer that future studies with a more thorough investigation are required to characterize the functional role of individual units, especially the controller units. We have edited the Discussion section to elaborate on this point.
> > > > >
> > > > > We agree that testing the robustness of timescale organization on models with different hyperparameters is important. Due to time limit we could not thoroughly examine the robustness of timescale organization results in LSTM with different hyperparameters. However, we have looked at whether timescale organization preserves in an LSTM with fewer hidden units (i.e., 100 hidden units instead of 650, still with 2 layers). We trained the model until perplexity was reduced to ~130 (again, due to time limits), and ran the same timescale and network analyses. (Please refer to the results figure here: https://anonymous.4open.science/repository/ef3696c5-e97e-4bd8-b12a-94795a038b8c/LSTM-100units.png ) We found that an LSTM with 100 units showed a similar timescale organization pattern as LSTM with 650 units, insofar as there was a smaller set of long-timescale units compared to short-timescale units. However, the longest timescale of LSTM with 100 units was shorter than the longest timescale for an LSTM with 650 units. Although we only trained this 100 unit-LSTM for a limited time, the validation error had already plateaued for several epochs. This may suggest that a smaller number of hidden units could limit the model capacity of learning long-timescale information. Again, we agree with the reviewer that this is an interesting and important question to be more thoroughly explored in the future.
> > > > >
> > > > > Also, for your previous question regarding Unit 823 in CLSTM, we have checked the activation difference curve of Unit 823 in a different context condition, where we segment the context and shared input in the middle of a sentence (at 100th character) instead of at the conjunction. We found that the sharp increase no longer exists (Please refer to the figure here: https://anonymous.4open.science/repository/ef3696c5-e97e-4bd8-b12a-94795a038b8c/unit823Check.png). Therefore, it is possible that the pattern is indeed due to this unit’s sensitivity to something like the “start of phrase” or “start of clause” or “syntactic head” information which could occur following the “, and” conjunction, as we previously speculated.

---

### Official Review · AnonReviewer2 · 2020-10-28
**Exploratory analyses of timescales in neural language models**

**Rating:** 6
**Confidence:** 4

**Review:**

This paper applies tools from neuroscience to understand how language models integrate across time. The basic approach is to present a phrase, preceded by two different context phrases: one that is natural (i.e. the phrase that actually preceded it in the corpus) and one that is randomly selected. The authors then measure how long it takes for the unit activations to become similar for the two different contexts, which provides a measure for how long the context impacts the representation. They find that (1) timescales increase at later layers of the language model (2) that only a small fraction of units exhibit long timescales (3) that long/medium-timescale units appear to come in two forms which they try and characterize using graph-style analyses.

--

Pros:

How language models integrate across time is clearly important, and this paper describes interesting first steps in characterizing the analysis of time using relevant tools from the neuroscience literature.

The method presented is simple and broadly applicable.

The graph-style results seem intriguing if a little hard to make sense of.  I also think that the sparsity of the long-timescale units is cool and interesting.

--

Limitations and questions:

1.	It’s not clear to me if the notion of time is a meaningful one in a language model. For example, the duration of contextual effects on a unit that codes syntactic number will presumably be highly variable and depend upon the details of the particular sentence being encoded. Thus a natural question is how variable are these timescales from moment-to-moment? What’s being plotted is the average across a bunch of sentences, segmented at a particular moment (a conjunction). How robust are these results if one examines a different point in a sentence? Are the timescales of some units more variable than others? -- Update: the authors have repeated their analysis for a different sentence point (after the 10th word) and report similar results. This analysis is helpful, though of course the 10th word is not a very principled break point, and there presumably is a lot of variation in timescales that are being averaged across. I continue to wonder how meaningful the notion of an absolute timescale is. --

2.	None of the steps in the graph analyses seemed particularly natural or well-motivated to me. Why were the graph edges thresholded at z>5 and why was k-core analysis performed? I find it hard to make sense of what this analysis tells us about how language information is processed. Is there some reason why medium timescale “controller” units and long-timescale “integrator” units should help with language processing? If these results are purely exploratory and lack a clear interpretation, then perhaps the authors could help the reader by explaining the thought process behind the exploration. Perhaps starting with the MDS plot would be useful rather than the k-core analysis, because the MDS plot clearly shows some interesting structure. -- The authors have motivated some of their analyses by discussing brain research reporting that longer-timescale regions are more densely connected. Of course, the relationship between connectivity between large-scale brain regions and the units in a LSTM remains highly speculative. But having some motivation is helpful. --

3.	It would be interesting to know how dependent these findings are on the model’s architecture. Would similar results be found for a Transformer or a simpler GRU-style RNN? -- The authors have attempted to address this point, but with limited time were not able to train a network to a high level of performance. --

--

Minor points:

In Figure 4, it would be helpful if the absolute timescale was labeled in all plots rather than the rank of the unit or the “normalized timescale”. The absolute timescale seems much more meaningful to me (and the units can of course still be ranked, just the axis labels changed or augmented).

The legend for Figure 4c is incorrect.

---

> ### Author Response · Authors · 2020-11-20
> **Official response to Reviewer 2**
>
> Thank you for your valuable feedback! Please see our point-by-point response below:
>
> - *It’s not clear to me if the notion of time is a meaningful one in a language model. For example, the duration of contextual effects on a unit that codes syntactic number will presumably be highly variable and depend upon the details of the particular sentence being encoded. Thus a natural question is how variable are these timescales from moment-to-moment? What’s being plotted is the average across a bunch of sentences, segmented at a particular moment (a conjunction). How robust are these results if one examines a different point in a sentence? Are the timescales of some units more variable than others?*
>
> Thank you for raising the point. Indeed, the timescale of individual units could vary based on the syntactic distance, which can be different from the token distance measured in our study. As pointed out by reviewer 3, one would expect that functional units such as syntax or number units should “reset” their timescale at certain points of the sentence (please refer to our response to the figures generated for Reviewer 3 regarding the reset process).
>
> Thank you for the suggestion about testing other locations in sentences, rather than just the specific “, and” conjunction. We conducted a new analysis in which we segmented the context and shared segments at a fixed distance from the sentence onset (i.e. we pick the 10th token as the segmentation point). We found that the timescale organization was largely preserved, regardless of whether we measured the cross-context effect at the “, and” boundary [as in our original analysis] or simply before and after the 10th token [as in this new analysis]. (Please refer to the figure here: https://anonymous.4open.science/repository/ef3696c5-e97e-4bd8-b12a-94795a038b8c/timescale_corr_commaAnd_middle.png).
>
> At the same time, we did find that there was a small subset of units (fewer than 10 units) whose timescales were clearly shorter following the “, and” conjunction. This is intriguing, as it suggests that some units were “resetting” their context following the “, and” conjunction. In future work, we will also seek to identify whether these units.
>
> In all, we agree with the reviewer that the timescales measured using “tokens” may be flexible, and can vary according to the syntactic context, for example.  To address how syntactic distance affects context encoding in individual unit, carefully controlled context conditions are needed, which is beyond the scope of the current study. However, we would like to stress the importance of token distance for RNNs in general, since it serves as an important parameter for RNNs to predict the next token in any kind of sequences, before it learns to flexibly integrate information based on syntactic distance in language specifically.
>
> We will revise the manuscript to include the analysis suggested by the reviewer, which suggests that the timescales measured at the level of tokens are robust across different choices of the location of the prior context segment.

---

> > ### Author Response · Authors · 2020-11-20
> > **Official response to Reviewer 2 (cont.)**
> >
> > - *None of the steps in the graph analyses seemed particularly natural or well-motivated to me. Why were the graph edges thresholded at z>5 and why was k-core analysis performed? I find it hard to make sense of what this analysis tells us about how language information is processed. Is there some reason why medium timescale “controller” units and long-timescale “integrator” units should help with language processing? If these results are purely exploratory and lack a clear interpretation, then perhaps the authors could help the reader by explaining the thought process behind the exploration. Perhaps starting with the MDS plot would be useful rather than the k-core analysis, because the MDS plot clearly shows some interesting structure.*
> >
> > Thank you for your comments. We agree with the reviewer that our analyses combining network connectivity with unit timescales are exploratory. We will revise the manuscript to make clearer that these analyses are motivated by a (general) hypothesis about the functional organization of the LSTM system, inspired by the hierarchical structure of the human brain.
> >
> > In the human brain, timescales of processing and anatomical connectivity are related:
> > More sensory regions (near the periphery of the cortical network) tend to have shorter timescales and lower degree; while higher-order regions (near the interior core of the cortical network) tend to have longer-timescales and higher degree.
> >
> > Moreover, human brain networks have a core periphery structure, in which a relatively small number of “higher order” and high-degree regions (in prefrontal cortex, in default-mode regions and in so-called “limbic cortex”) maintain a large number of connections with one another, and exert a powerful controlling role over large-scale cortical dynamics. This notion of a network core is longstanding – see Figure 2 in Mesulam (1998) (https://doi.org/10.1093/brain/121.6.1013) or Figure 5 in Hagmann et al. (2008) (https://doi.org/10.1371/journal.pbio.0060159), for variations of this idea.
> >
> > Therefore, we were interested to test the (tentative) hypothesis that higher degree nodes in LSTMs might have longer timescales, and lower degree nodes might have shorter timescales. This is the why we pursue these particular sets of network analyses, examining the k-cores, coupling maps, and their relationship to the timescales. Since language processing involves lower to higher-level cortices in the brain, understanding the similarities/differences of functional organization between the brain and language models should help us understand how language information is processed in language models.
> >
> > What do the “integrator” and “controller” units tell us about language processing? As this is our first effort to construct the functional maps, it is too early to make specific statements. We feel that the more important contribution is that the mapping tools applied here enable us to narrow down the range of candidate nodes that could be in involved in long-range dependency tracking, which presumably is associated with more high-level grammatical and discourse level-processing. The aim of this mapping procedure is to serve as a guide to future, more targeted investigations.
> >
> > In terms of the details of how we chose the threshold, we would like to clarify that the threshold |z| > 5 was only used to investigate the relationship between number of strong projections and the timescale of individual units (shown in Figure 3A), but not used to construct graph edges. Furthermore,  the threshold does not really affect the current results that units with longer-timescales tend to have more strong projections: we observed significant correlation between timescale and number of strong projections using multiple thresholds:|z|>3 (r=0.3), |z|>4 (r = 0.35) and |z|>5 (r =0.29). Also, we got significant correlations when varying the thresholds in the character-level LSTM model (|z|>5, r = 0.24; |z|>4, r = 0.30; |z|>3, r = 0.36)
> >
> > On the other hand, the graph edges for constructing the network shown in Figure 3C were identified using the top 258 weight values in the LSTM hidden-to-gate weight matrix. These weights were not thresholded.
> >
> > Finally, thank you for the suggestions regarding the organization of the results. Since the interesting patterns revealed by the MDS come from the distinctive positions of controller and integrator units, we think it might be easier for the readers if we explain how the controller units were identified first. However, we completely understand your concern and will revise the manuscript to describe our motivation and methods regarding the k-core analysis more clearly.

---

> > > ### Author Response · Authors · 2020-11-20
> > > **Official response to Reviewer 2 (cont.)**
> > >
> > > - *It would be interesting to know how dependent these findings are on the model’s architecture. Would similar results be found for a Transformer or a simpler GRU-style RNN?*
> > >
> > > Thank you for the suggestion. We agree it would be interesting to use this model-free approach to explore the timescale organization of language models with a different architecture.
> > >
> > > To explore how architecture may influence the findings, we trained a word-level GRU language model using the same settings as used in Gulordava et al., including the same Wikipedia training corpus, the same loss function (i.e. cross-entropy loss), and the same hyperparameters. Specifically, the GRU model also has two layers with 650 hidden units in each layer as the LSTM model we analyzed in the current study.
> > >
> > > Unfortunately, due to limitations of time and computational resources, we had to stop training the model after 48 hours, at which point the GRU achieved a test perplexity of 349.39. This perplexity is much higher than the LSTM model reported in Gulordava et al. (perplexity = 52.1 in the English corpora) because they selected the best models after training for 40 epochs, whereas our testing GRU model was only trained for ~10.5 epochs.
> > >
> > > We then analyzed the timescale of hidden units using the same method as was used for analyzing the LSTMs, and using the test data derived from the training Wikipedia corpus (Please refer to the result figure here: https://anonymous.4open.science/repository/ef3696c5-e97e-4bd8-b12a-94795a038b8c/GRU_timescale_wikitest.png).
> > >
> > > Overall, we found that the majority of the units in the GRU also showed shorter timescales, similar to the Gulordova LSTM model, but that the relationship between timescales of each GRU unit and its projection patterns appeared different to the relationship observed for the LSTM. In particular: (1) similar to the word level LSTM, the second layer of the GRU model was more sensitive to prior context than the first layer. (2) The overall distribution of timescales in the GRU was similar to LSTM, although the GRU showed a right-skewed distribution with a larger proportion of short-timescale units relative to the word-level LSTM (Panel B and C). Of course, the differences between GRU and LSTM models may arise because of the limited training time for the GRU, and so can only be taken as provisional.
> > >
> > > We also performed the timescale vs. network connectivity analyses on this GRU model (Please refer to result figures here: https://anonymous.4open.science/repository/ef3696c5-e97e-4bd8-b12a-94795a038b8c/GRU_connectivity.png). Because the update of hidden states in GRU are controlled by the update gate, we chose to analyze hidden-to-update gate weight matrix. In contrast to the LSTM models, the GRU that we trained did not show the pattern that units with longer timescales exhibited more strong projections (Panel A). Moreover, when using k-core analysis to identify subunits of interconnected high-degree units, the core network contained many units with long to short timescales. Interestingly, when we visualize the position of the k-core units in the MDS space, they tended to locate at the edge of the space, similar to what we found in LSTM. This indicates that the k-core units have distinctive profiles, distant from one another and from other units in the network.
> > >
> > > Although the similarities/differences between LSTM and GRU here are intriguing, we should keep in mind that (1) the perplexity of this GRU is much higher than the LSTM, due to limitations in training time and tuning, and that (2) comparing the LSTM and GRU connection patterns is not straightforward, as the overall distribution of weights is different, so further work is required to determine comparable thresholds for “strong” projections and “high-degree units” in each case. As we noted in the manuscript and above, the connectivity results are exploratory; however, we believe that the GRU analysis shows how these methods can be extended to map and compare the functional organization of language models of different architectures.

---

> > > > ### Author Response · Authors · 2020-11-20
> > > > **Official response to Reviewer 2 (cont.)**
> > > >
> > > > - *Minor points: In Figure 3, it would be helpful if the absolute timescale was labeled in all plots rather than the rank of the unit or the “normalized timescale”. The absolute timescale seems much more meaningful to me (and the units can of course still be ranked, just the axis labels changed or augmented).*
> > > >
> > > > Because our interpolation FWHM method can only map timescales as an integer number of words, it would be difficult to visualize the exact timescale for every unit in Figure 3A : many units have the same timescale and would thus overlap visually. We will add the scatter plot of timescale variation as shown in: https://anonymous.4open.science/repository/ef3696c5-e97e-4bd8-b12a-94795a038b8c/timescale_corr_commaAnd_middle.png into Appendix for readers who would like to see the exact timescale of individual units.
> > > >
> > > > - *I thought the information in Appendix A.1 that describes the basic stimuli used should be added to the main text.*
> > > >
> > > > Thank you for your suggestion. We will add this information into main text.
> > > >
> > > > - *What matrix was MDS performed on? The thresholded connection matrix?*
> > > >
> > > > The matrix the MDS was performed on is the hidden-to-gate projection patterns which was obtained from the original weight matrices in LSTM (concatenating W_hi and W_hf ).
> > > >
> > > > - *Figure 3C didn’t seem to convey any useful information. Am I missing something?*
> > > >
> > > > Thank you for the comment. We will move this figure to Appendix.
> > > >
> > > > - *Why are there two bias terms in equations 4 and 5?*
> > > >
> > > > Thank you for pointing this out. We will revise the manuscript.
> > > >
> > > > - *Equations 4 and 5 have do not have the variable W_i and W_h mentioned above, only W_ii, W_if, W_hi, W_hf. I personally think it would be clearer to give the weights for the inputs and hidden units different names rather than different subscripts.*
> > > >
> > > > Thank you for the suggestion. We agree that it would be clearer to use different names and we will revise the manuscript accordingly.
> > > >
> > > > - *In Fig 1B, having a panel that plots the difference between intact and random context, might make the point clearer. Presumably this would show that the differences become smaller over time which is not obvious from the plot.*
> > > >
> > > > Thank you for your suggestion.  We will revise Figure 1 to make this clearer.

---

### Official Review · AnonReviewer4 · 2020-10-28
**Towards understanding the internal organisation of LSTMs**

**Rating:** 7
**Confidence:** 3

**Review:**

This paper looks at LSTMs with the intention of understanding their functional connectivity. I am not sure exactly what the relationship between the brain and LSTMs is being assumed or proposed herein — however I understand the need to understand complex neural networks regardless of their relationship to biological systems.

I would have liked to have a discussion with respect to what the hierarchical organisation is due to. Is this merely a repercussion of the connectivity, for example? What do the authors think?

In terms of work that looks at ablation (i.e., damage), it might be useful to bear in mind limitations of such work if various (seemingly, perhaps) extraneous factors are not taken into account, see: https://doi.org/10.1007/s42113-020-00081-z

I think this paper can be polished to the level of a solidly good paper if the authors can sketch out a bit more their rationale and syllogisms with respect to my above questions.

Minor:
* Figures are very hard to read, is it possible to redesign them slightly to make the text bigger?
* In LaTeX to open double quotes you need to use two backticks. Also the \cite and \citep commands should be used appropriately in terms of places where \citep is needed as well as use of optional arguments to avoid double parentheses.

---

> ### Author Response · Authors · 2020-11-20
> **Official response to Reviewer 4**
>
> First of all, thank you for your positive comments on the paper. Please see our response to the specific questions/comments below:
>
> - *I would have liked to have a discussion with respect to what the hierarchical organisation is due to. Is this merely a repercussion of the connectivity, for example? What do the authors think?*
>
> Is the hierarchical timescale phenomenon a direct result of the connectivity? At the level of layers (i.e. the fact that Layer 1 showed a shorter timescale overall, relative to Layer 2), this effect is likely due to connectivity. If unit B in Layer 2 receives an input from unit A in Layer 1, and if Unit A is sensitive to changes in the input from N words earlier, then that context-sensitive activation is passed as an input to Unit B. As a result, Unit B will most likely show at least some sensitivity to changes in the input from N words earlier. Thus, on average, when an entire population of units is downstream from another population, we should expect the downstream layer to have longer timescales, on average.
>
> The link to connectivity is less straightforward for within-layer connections. Models from the neuroscience literature indicate that higher-degree nodes in dynamical systems will (under some conditions) tend to exhibit slower dynamics than lower-degree nodes (Baria et al., 2013,  https://doi.org/10.1016/j.neuroimage.2013.01.072), so one might expect that higher-degree nodes (simply by virtue of changing state more slowly) should also have longer context-dependence. However, this is not always the case, as shown in the GRU model that we trained to test the generality of the timescale-connectivity findings (Please see our Response to Reviewer 2). Moreover, it is certainly theoretically possible for a small subset of very long-timescale nodes to operate through a connection bottleneck.
>
> We will revise the manuscript to note these connectivity-timescale relationships, which were also of interest to Reviewer 2.
>
> -  *In terms of work that looks at ablation (i.e., damage), it might be useful to bear in mind limitations of such work if various (seemingly, perhaps) extraneous factors are not taken into account, see: https://doi.org/10.1007/s42113-020-00081-z*
>
> Thank you for sharing the interesting article. We agree that ablation methods have some limitations and therefore in the current study we proposed this model-free method to investigate the functional property of individual units without lesioning the model. We will cite this paper in relation to our own new ablation findings in the revised paper.
>
>
> - *Minor:
> Figures are very hard to read, is it possible to redesign them slightly to make the text bigger?*
>
> We will get feedback from colleagues and attempt to improve the readability and scaling of the figure panels.
>
>
> - *In LaTeX to open double quotes you need to use two backticks. Also the \cite and \citep commands should be used appropriately in terms of places where \citep is needed as well as use of optional arguments to avoid double parentheses.*
>
> We will revise the quotation and citation format. Thank you for the tips!

---

### Decision · Program_Chairs · 2021-01-07
**Final Decision**

**Decision:**

Accept (Poster)

**Comment:**

This paper applies methods inspired by neuroscience to analyze the inner workings of LSTM language models. In particular, a simple and clever approach is proposed, in which a sentence is presented in its observed context vs. a random one. The time for a unit activation to become similar in the two contexts is used as a probe of the timescale of contextual effects. The main results are that timescales increase with layer and that there are two classes of long-timescale units with different graph-theoretical properties. The functionality of syntax-sensitive units previously identified in the literature is confirmed. Finally, the analysis is replicated for a character-level model.

The paper received detailed and insightful reviews, and there was a lively (but always respectful) discussion between authors and reviewers.

Overall, the reviewers liked the topic of the paper and the overall methodology, however they had several issues with it. One of the issue pertained to the "holistic" approach to time in the paper, which is measured in number of tokens, rather than in terms of syntactic distance. More in general, there was a feeling that the paper was somewhat short on actual insights on the exact functional role of units in a linguistic context. The reviewer who assigned the most severe score was mostly concerned about one specific instance of this, namely the fact that the authors focus on syntax-tracking and number agreement units whose scope should not really extend across sentences. Moreover, the reviewer was surprised that the syntax-tracking units maintain information across longer distances than the number-agreement units, that should, by definition, keep track of long-distance relations.

I am divided. I welcome work that focuses on novel qualitative and quantitative analyses of an existing model. I wished there were clearer take-home messages on how LSTMs process language, but I recognize that our knowledge of deep-learning models is very preliminary, and I am thus not surprised that the conclusions are not entirely clear.  The reviewers raised important concerns, but I would not confidently claim that we know enough about the relevant units to be genuinely surprised by some of the results. For example, can we really say that number-agreement units are only limited to clause-internal agreement tracking? Couldn't it be, say, that we will discover in the future they also play a role in tracking discourse-determined pronominal number (going out on a random limb, here, of course)?

Overall, I would like to see this at least as a poster at the conference, but I am assigning low confidence to my recommendation as I respect the reviewers' point of view.